# Ortholog of autism candidate gene *RBM27* regulates mitoribosomal assembly factor MALS-1 to protect against mitochondrial dysfunction and axon degeneration during neurodevelopment

Tamjid A. Chowdhury[1], David A. Luy[1], Garrett Scapellato[1], Dorian Farache[2,3], Amy S. Y. Lee[2,3], Christopher C. Quinn[1]*

1 Department of Biological Sciences, University of Wisconsin-Milwaukee, Milwaukee, Wisconsin, United States of America, 2 Department of Cell Biology, Harvard Medical School, Boston, Massachusetts, United States of America, 3 Department of Cancer Immunology and Virology, Dana-Farber Cancer Institute, Boston, Massachusetts, United States of America

* quinnc@uwm.edu

## Abstract

Mitochondrial dysfunction is thought to be a key component of neurodevelopmental disorders such as autism, intellectual disability, and attention-deficit hyperactivity disorder (ADHD). However, little is known about the molecular mechanisms that protect against mitochondrial dysfunction during neurodevelopment. Here, we address this question through the investigation of *rbm-26*, the *Caenorhabditis elegans* ortholog of the *RBM27* autism candidate gene, which encodes an RNA-binding protein whose role in neurons is unknown. We report that RBM-26 (RBM26/27) protects against axonal defects by negatively regulating expression of the MALS-1 (MALSU1) mitoribosomal assembly factor. Autism-associated missense variants in RBM-26 cause a sharp decrease in RBM-26 protein expression along with defects in axon overlap and axon degeneration that occurs during larval development. Using a biochemical screen, we identified the mRNA for the MALS-1 mitoribosomal assembly factor as a binding partner for RBM-26. Loss of RBM-26 function causes a dramatic overexpression of *mals-1* mRNA and MALS-1 protein. Moreover, genetic analysis indicates that this overexpression of MALS-1 is responsible for the mitochondrial and axon degeneration defects in *rbm-26* mutants. These observations reveal a mechanism that regulates expression of a mitoribosomal assembly factor to protect against axon degeneration during neurodevelopment.

## Introduction

Emerging evidence suggests an association between mitochondria and neurodevelopmental disorders including autism, intellectual disability, and attention-deficit hyperactivity disorder (ADHD). For example, certain mitochondrial DNA lineages confer substantial risk for autism

**Data Availability Statement:** All relevant data are within the paper and its Supporting information files.

**Funding:** This work was funded by the National Institute of Mental Health grant R01MH119157 (to CCQ), the National Institute of Neurological Disorders and Stroke grant R03NS101524 (to CCQ), Discovery and Innovation Grant (101X432) from the University of Wisconsin - Milwaukee (to TAC) and National Institutes of Health grant 1R35GM142527 (to ASYL). Some strains were provided by the Caenorhabditis Genetics Center (funded by NIH P40 OD010440). This article does not represent the official views of the National Institutes of Health and the authors bear sole responsibility for its content. The funders had no role in study design, data collection and analysis, decision to publish, or preparation of the manuscript. The URL for the NIH is: https://www.nih.gov/. The URL for University of Wisconsin - Milwaukee is: https://uwm.edu/.

**Competing interests:** The authors have declared that no competing interests exist.

**Abbreviations:** ADHD, attention-deficit hyperactivity disorder; AID, auxin-induced degron; HDR, homology directed repair; NGM, nematode growth medium; RMCE, recombinase mediated cassette exchange; ROI, region of interest; ROS, reactive oxygen species; SFARI, Simons Foundation Autism Research Initiative; SMA, spinal muscular atrophy.

and ADHD [1,2]. Likewise, heteroplasmic mtDNA mutations are also a risk factor for autism and intellectual disability [3–5]. In mice, a hypomorphic missense mutation in the *ND6* mtDNA gene exhibit autism endophenotypes and also produce excess reactive oxygen species (ROS) in the brain, suggesting that mitochondrial dysfunction is causative for autism-like phenotypes in mice [6]. In this regard, it is intriguing to note that elevated levels of ROS have also been observed in autistic human brains [7]. Together, these observations suggest that mitochondrial dysfunction might play a causative role in neurodevelopmental disorders. However, the roles of mitochondrial dysfunction in neurodevelopmental disorders are not well understood.

Some neurodevelopmental syndromes feature neurodegenerative phenotypes that begin during the developmental time period. For example, loss of *EXOSC3* or *EXOSC9* gene function has been associated with a neurodevelopmental syndrome that includes intellectual disability and axon degeneration during infancy [8–10]. Likewise, loss of the *GAN* gene in humans has been associated with giant axonal neuropathy disease, featuring intellectual disability and axon degeneration in children [11]. Moreover, loss of either the *SPTBN1* or *ADD1* genes have been associated with a neurodevelopmental syndrome that includes autism and ADHD [12,13]. In mouse models, variants in these genes cause defects in axon development along with axon degeneration, suggesting that loss of these genes might be causative for the human syndrome [14,15]. Together, these observations suggest that neurodegeneration may contribute to some neurodevelopmental disorders. However, little is known about the cellular and molecular mechanisms that protect against neurodegeneration during neurodevelopment.

The *RBM27* (*rbm-26*/*Swm*/*Rmn1*) gene encodes an RNA-binding protein and is considered an autism candidate gene by the Simons Foundation Autism Research Initiative (SFARI) [16]. In a cultured human cell line, RBM27 and its paralog RBM26 can function with the PAXT connection, an adaptor complex for the nuclear RNA exosome [17]. Likewise, in *S. pombe* the Rmn1 ortholog of RBM26/27 can also function as part of an adaptor for the nuclear RNA exosome [18]. In *Drosophila*, the Swm ortholog of RBM26/27 has also been implicated in RNA metabolism and the maintenance of adult intestinal stem cells [19]. Despite these findings, the role of RBM27 and its orthologs in neurons have not been investigated. Moreover, the identity of *RBM27* as an autism candidate gene is based on de novo missense mutations that have not been tested for gene-disrupting activity.

Here, we identify a mitochondria-protecting role for the RBM-26 ortholog of RBM26/27 that protects mitochondria and prevents axon degeneration during larval development. We find that loss of RBM-26 function causes a decrease in mitochondrial density in the axon, mitochondrial dysfunction in the cell body, axon degeneration, and transient axon overlap defects. These phenotypes are induced by an *rbm-26* null allele as well as by alleles that are equivalent to autism-associated de novo missense variants in *RBM27*, suggesting that these are likely gene-disrupting variants in humans. Mechanistically, we identified the *mals-1* mRNA as a binding partner for RBM-26 and found that RBM-26 negatively regulates the expression of the *mals-1* mRNA, which encodes a mitoribosomal assembly factor. Moreover, gain-of-function and loss-of-function experiments indicate that MALS-1 is required for both the axon degeneration phenotype and the reduction in axonal mitochondria that are caused by loss of RBM-26. These observations indicate that RBM-26 negatively regulates MALS-1 expression to protect against mitochondrial dysfunction and axon degeneration during development.

## Results

### Expression of RBM-26 is disrupted by P80L and L13V missense mutations

*RBM27* has been identified as a candidate autism gene by the SFARI [16]. The potential association of *RBM27* with autism is based on the identification of 5 de novo variants in probands

with autism or other neurodevelopmental disorders, but none in unaffected siblings (Fig 1A) [20–25]. However, the confidence in this association is limited by the fact that the identified variants are all missense variants, rather than likely-gene-disrupting variants. Moreover, a role for *RBM27* in neurodevelopment has not been investigated. To address these issues, we used the *Caenorhabditis elegans rbm-26* ortholog of *RBM27*. We found that 2 of the 5 de novo missense mutation sites in human RBM27 are conserved in *C. elegans rbm-26*. These 2 conserved missense mutations cause L13V and P79L amino acid changes in human RBM27, which are equivalent to L13V and P80L respectively in *C. elegans* RBM-26 (Fig 1B). Both of these mutations occur in the PWI-like domains and were identified in genome sequencing studies of children with neurodevelopmental disorders [20,23].

To determine how the L13V and P80L variants affect RBM-26 protein expression, we used CRISPR to introduce these mutations into the endogenous *rbm-26* gene and also added a 3XFLAG tag to allow for detection by western blot. A sequence encoding 3XFLAG was inserted just upstream of the *rbm-26* stop codon, thereby creating the *rbm-26(cue25)* allele, hereafter referred to as *rbm-26(3XFLAG)*. Next, we used CRISPR to introduce the P80L mutation into the *rbm-26(3XFLAG)* allele, thereby creating the *rbm-26(cue22cue25)* allele, hereafter called *rbm-26(P80L::3XFLAG)*. We also used CRISPR to introduce the L13V mutation into the *rbm-26(3XFLAG)* allele, thereby creating the *rbm-26(cue24cue25)* allele, hereafter called *rbm-26(L13V::3XFLAG)*.

We next used western blotting to detect the RBM-26::3XFLAG, RBM-26 P80L::3XFLAG, and RBM-26 L13V::3XFLAG proteins (Figs 1C and S1). The wild-type RBM-26::3XFLAG protein is predicted to produce proteins with molecular weights around 75 and 82 KD. Consistent with this prediction, we found that the RBM-26::3XFLAG protein was detected as 2 bands that migrate in between the 72 KD and 95 KD markers. In addition, we also observed a more intense bands just above the 56 KD marker and just above the 95 KD marker as well as 2 fainter bands in between the 34 KD and 56 KD markers. The identities of these additional bands are unknown, though they are likely the result of posttranslational processing, or unpredicted splice forms. We found that each of these RBM-26::3XFLAG bands were sharply reduced by the P80L and L13V mutations (Fig 1C and 1D). To determine if the P80L mutation can reduce expression of RBM-26 in vivo, we used an endogenously tagged *rbm-26::scarlet* gene (Fig 1E and 1F). Consistent with our western blotting data, we found that the P80L mutation substantially reduces expression of the RBM-26::Scarlet protein in vivo. These observations indicate that the P80L and L13V mutations in *C. elegans rbm-26* sharply reduce RBM-26 expression and imply that the P79L and L13V mutations in human *RBM27* are likely to be gene-disrupting.

## Mutations in *rbm-26* cause a transient ALM/PLM axon overlap defect

To determine if RBM-26 can affect neuronal development, we used the PLM, a neuron that has its cell body in the tail and extends its axon anteriorly along the body wall. In late larval stages and adults, the PLM axon terminates prior to the ALM neuron [26]. The PLM axon grows in 3 phases [26]. The first phase occurs during the first part of the L1 stage when the axon grows rapidly and overshoots the ALM neuron, thereby creating a transient PLM/ALM overlap. The second phase occurs during the later L1 stage, when PLM axon growth pauses, allowing for the resolution of the PLM/ALM overlap. The third phase of PLM axon growth occurs throughout the remainder of larval development and features continuing axon growth that matches the speed of body growth.

To determine how loss of RBM-26 function affects the PLM, we used an *rbm-26(null)* mutation, along with *rbm-26(P80L)* and *rbm-26(L13V)* mutations. The *rbm-26(gk910)* null

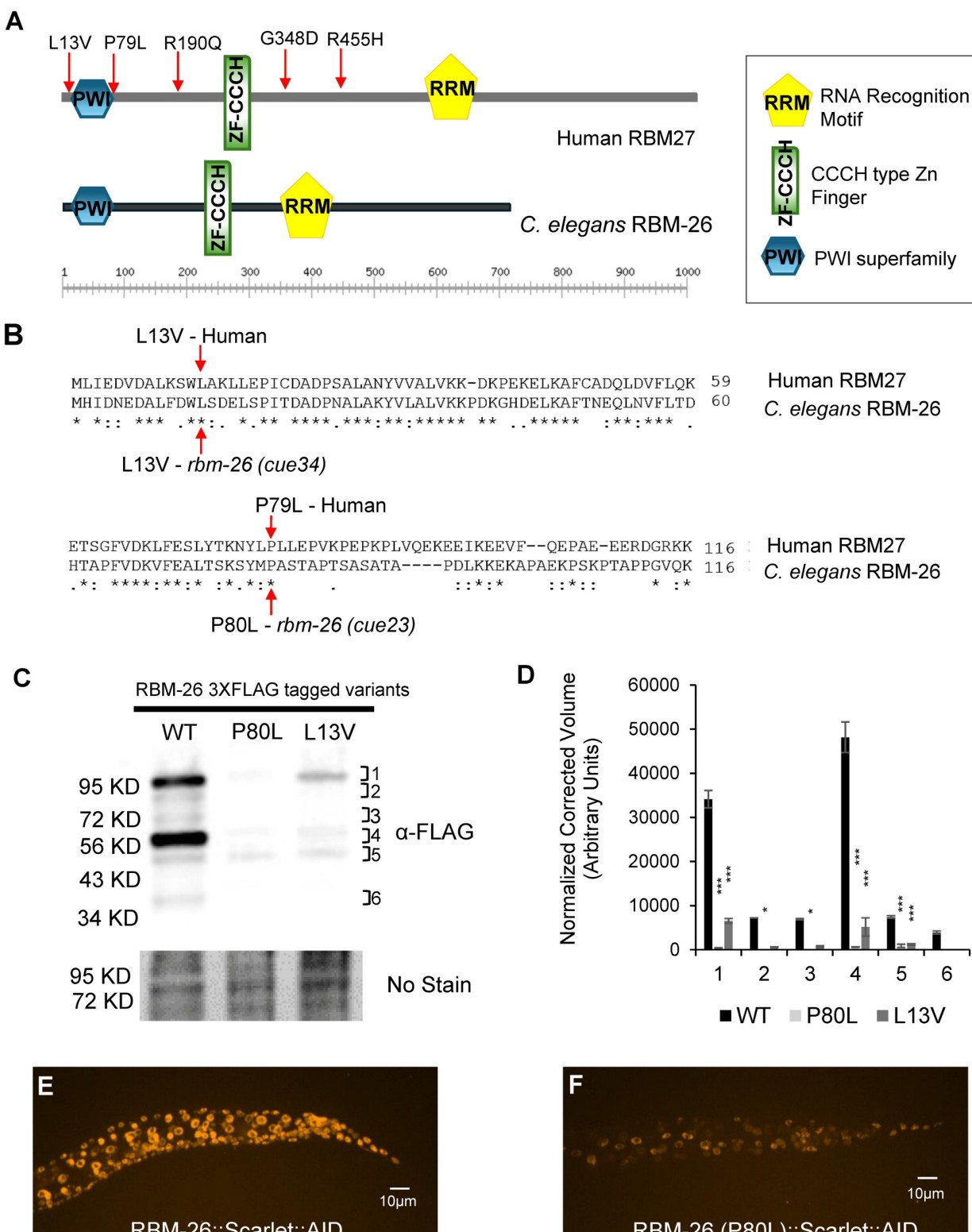

**Fig 1. Expression of RBM-26 is disrupted by P80L and L13V missense mutations.** (A) Putative protein domains in human RBM27, and *C. elegans* RBM-26 proteins predicted by NCBI's CD-Search. CCCH-type Zn Finger domains (Green Rectangles) and RNA Recognition Motifs (Gold Pentagons) were predicted with high confidence by NCBI's CD-Search and SMART database. PWI superfamily (Blue Hexagon) was predicted by NCBI's CD-Search with high confidence. Red arrows indicate de novo missense variants identified in patients with autism or other

neurodevelopmental disorders. (B) Clustal Omega alignment of the N-terminus of the human RBM27 and *C. elegans* RBM-26 proteins. Two de novo RBM27 variants, L13V and P79L (red arrows), were found in regions conserved in *C. elegans* RBM-26. (C) Representative western blot of 3 biological replicates showing expression of RBM-26::3XFLAG, RBM-26 P80L::3XFLAG, and RBM-26 L13V::3XFLAG proteins; 20 μg of total protein lysate was loaded per well and specific proteins were detected with an anti-FLAG antibody and enhanced chemiluminescence. Protein loading was quantified by No-Stain Protein Labeling Reagent. (D) Densitometric quantification of protein bands. Statistical significance was analyzed by one-way ANOVA with multiple comparison testing (*$p < 0.05$, ***$p < 0.0001$). (E) Example of endogenously tagged RBM-26 in wild type at the L3 stage. (F) Example of endogenously tagged RBM-26 in *rbm-26(P80L)* mutant at the L3 stage. RBM-26 was endogenously tagged with Scarlet using CRISPR to generate the *rbm-26(syb2552[rbm-26::Scarlet::AID]) allele*. CRISPR was used to insert the P80L mutation into *rbm-26(syb2552)* to create the *rbm-26(cue48[rbm-26(P80L)] syb2552[rbm-26::Scarlet::AID])* allele. Underlying data can be found in S1 Data.

mutation consists of a 408 base pair deletion that removes part of exon 4 and exon 5 of *rbm-26* and is lethal. We maintained this *rbm-26(null)* allele over a balancer and found that maternally rescued homozygote progeny are viable until the L3 stage. To study the P80L missense mutation, we used CRISPR to create the *rbm-26(cue23)* allele, hereafter referred to as *rbm-26(P80L)*. We also introduced the L13V missense mutation into the *rbm-26* gene, creating the *rbm-26(cue34)* mutation, hereafter referred to as *rbm-26(L13V)*. In contrast to the *rbm-26(null)* mutation, both the *rbm-26(P80L)* and the *rbm-26(L13V)* mutations are viable.

We found that mutations in *rbm-26* cause overlap between the PLM and ALM axons, hereafter referred to as the PLM/ALM overlap phenotype (Fig 2A–2D). In *rbm-26(null)* mutants, a greater penetrance of PLM/ALM overlap was observed in the L1, L2, and L3 stages relative to wild type. Due to the lethal phenotype of the *rbm-26(null)* mutation, we were unable to analyze PLM axons in in later stages. In the P80L and L13V mutants, we also observed a greater penetrance of PLM/ALM overlap in the L2 and L3 stages relative to wild type. However, we did not observe PLM/ALM overlap at later stages in these missense mutations, suggesting that the PLM/ALM overlap defect resolves at later developmental stages (Fig 2D). Analysis of the PLM axon length to body length ratio suggests that the PLM/ALM overlap phenotype in these 3 *rbm-26* mutants is caused by overextension of the PLM axon (S2 Fig). Taken together, these observations suggest that loss of RBM-26 function might cause a delay in the refinement of ALM/PLM overlap.

## Mutations in *rbm-26* cause PLM axon degeneration phenotypes that begin during larval stages

We found that the *rbm-26(null)*, *rbm-26(P80L)*, and the *rbm-26(L13V)* mutations all cause degeneration phenotypes in the PLM axon (Figs 2E–2H and S3 and S1 Table). In the *rbm-26(null)* mutants, we observed a very weak axon beading phenotype in the L2 stage that became much stronger in the L3 stage (Fig 2F). The *rbm-26(P80L)* mutation caused an axon beading phenotype that was very weak in the L2 stage, increased in the L3 and L4 stages and became much stronger in the adult stage. We also observed the beading phenotype in *rbm-26(L13V)* mutants, but at a substantially lower penetrance relative to *rbm-26(P80L)*. In addition to beading, we observed several other more subtle degeneration phenotypes (S1 Table), including axon breaks (Fig 2G and 2H). To determine if loss of *rbm-26* function might cause defects in other neurons, we also observed the PVD neuron and found subtle signs of degeneration in both its axon and dendrites (S4 Fig). These degenerative phenotypes in PLM and PVD are similar to those that have been reported in aging worms [27–29] and are reminiscent of axonopathy that has been reported in human neurodegenerative disorders and described as axonal swellings or spheroids [30–32]. Overall, these observations suggest that RBM-26 protects against axon degeneration. Moreover, these results imply that the P79L and L13V de novo missense mutations in human *RBM27* are likely to disrupt *RBM27* function, thereby supporting a role for this gene in neurodevelopmental disorders.

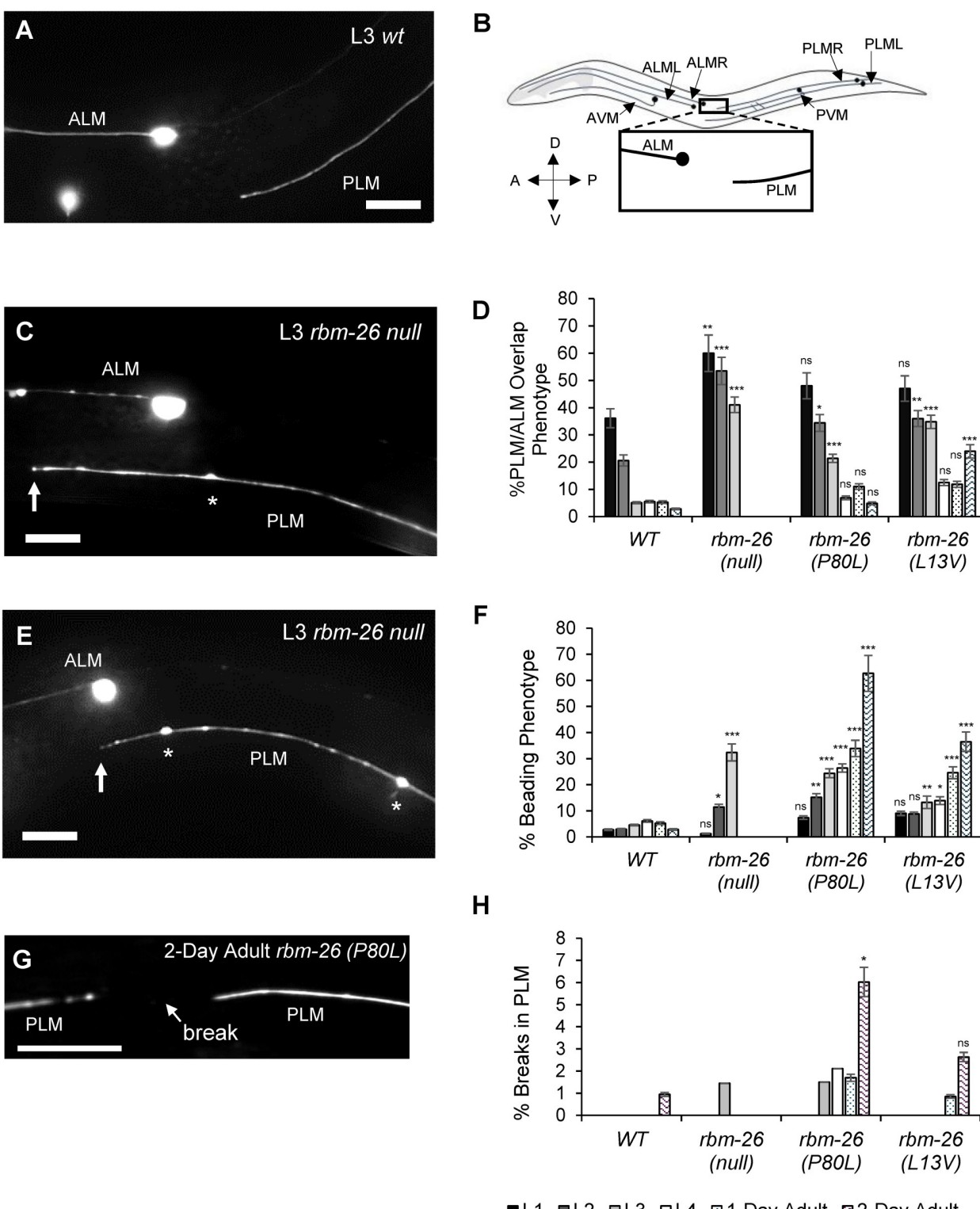

**Fig 2. Mutations in *rbm-26* cause axon degeneration and axon overlap defects.** (A) Example of a normal PLM axon, where the PLM axon does not overlap with the ALM axon (B) Diagram of the touch receptor neurons. (C) Example of an axon overlap defect (PLM/ALM overlap phenotype) in *rbm-26 (null)* L3 worms. Axon beading can also be seen (asterisk). (D) Quantification of axon overlap defects in various stages. (E) Example of axon beading phenotype (asterisks) in an *rbm-26(null)* mutant. (F) Quantification of beading phenotype at various stages. (G) Example of the axonal break phenotype in the PLM of a 2-day-old adult *rbm-26 (P80L)* mutant. (H) Quantification of the axonal break phenotype at various stages.

Asterisks indicate statistically significant difference relative to wild type, Z-test for proportions (*** $p < 0.0001$, ** $p < 0.01$, and * $p < 0.05$), and error bars represent the standard error of the proportion. Scale bars are 10 μm. Axons are visualized with the *muIs32 (Pmec-7::gfp)* transgene. Stages are labeled with L1 (first larval stage), L2 (second larval stage), L3 (third larval stage), L4 (fourth larval stage), 1-Day Adult and 2-Day Adult. Alleles: *rbm-26(null)* is *rbm-26(gk910)*; *rbm-26(P80L)* is *rbm-26(cue23)*; *rbm-26(L13V)* is *rbm-26(cue34)*. Underlying data can be found in S1 Data.

## RBM-26 is expressed in multiple tissues and functions within neurons to protect against axonal defects

To visualize the localization of RBM-26, we used CRISPR to insert the coding sequence for a Scarlet::AID (auxin-inducible degron) tag into the 3′ end of the *rbm-26* gene, thereby creating the *rbm-26(syb2552)* allele. We found that this allele expresses RBM-26::Scarlet::AID protein in many tissues including neurons, hypodermis, muscle, and intestine (Figs 3A–3C, S5 and S6). Within hypodermis and neurons, we noticed that the RBM-26::Scarlet::AID protein was localized to the nucleus (Figs 3A–3C and S5).

We next asked where RBM-26 functions to protect against axon degeneration and axon overlap defects. For our initial approach to this question, we used the auxin-inducible degron system to selectively degrade the RBM-26::Scarlet::AID protein in neurons, hypodermis, muscle, and intestine [33]. We found that degradation of RBM-26::Scarlet::AID in neurons caused both PLM/ALM overlap defects (Fig 3D) and the PLM beading phenotype (Fig 3E). However, degradation of RBM-26::Scarlet::AID in hypodermis, muscle, or intestine failed to cause any of these phenotypes (Fig 3D and 3E). These phenotypes were dependent on auxin and the *rbm-26::scarlet::AID* gene (S7 and S8 Figs). We also note that both of these phenotypes occurred at a lower penetrance in these degradation experiments relative to experiments with the *rbm-26* mutant alleles, suggesting incomplete degradation of the RBM-26::Scarlet::AID protein.

To further test the locus of RBM-26 function, we conducted transgenic rescue experiments. For the transgenic rescue experiments, we used a *Pmec-7::rbm-26* transgene to selectively drive expression of RBM-26 in the touch receptor neurons (Fig 3F and 3G). We found that this *Pmec-7::rbm-26* transgene could rescue the PLM/ALM axon overlap phenotype in both the *rbm-26(P80L)* mutants and in the *rbm-26(L13V)* mutants (Fig 3F). Likewise, the *Pmec-7::rbm-26* transgene could also rescue the beading phenotype in both the *rbm-26(P80L)* mutants and in the *rbm-26(L13V)* mutants (Fig 3G). We note that overexpression of RBM-26::Scarlet also caused a low penetrance of PLM/ALM axon overlap defects, suggesting that excessive function of RBM-26 may also cause some axon overlap defects. These observations suggest that RBM-26 is broadly expressed and that it functions cell-autonomously within neurons to protect against axon degeneration and axon overlap defects. However, given the widespread expression of RBM-26, we cannot exclude the possibility that it might also have non-autonomous functions.

## Loss of RBM-26 causes mitochondrial dysfunction

Mitochondrial dysfunction has been associated with neuronal degeneration. Therefore, we asked if loss of RBM-26 function affects mitochondria in the PLM. We used the third larval (L3) stage for these experiments, because the *rbm-26(null)* mutants die after this stage. We found that mitochondria were localized throughout wild-type PLM axons, with an average density of 9 per 100 μm (Fig 4A, 4B and 4E). The *rbm-26(null)* mutation caused a reduction of the mitochondrial density in the PLM axon to about 6 per 100 μm (Fig 4C–4E). Likewise, the *rbm-26(P80L)* and the *rbm-26(L13V)* mutations also caused a reduction of the mitochondrial density in the PLM axon (Fig 4E). To determine the timing of this phenotype, we analyzed *rbm-26(null)* mutants at earlier larval stages and found that the phenotype begins in the L1

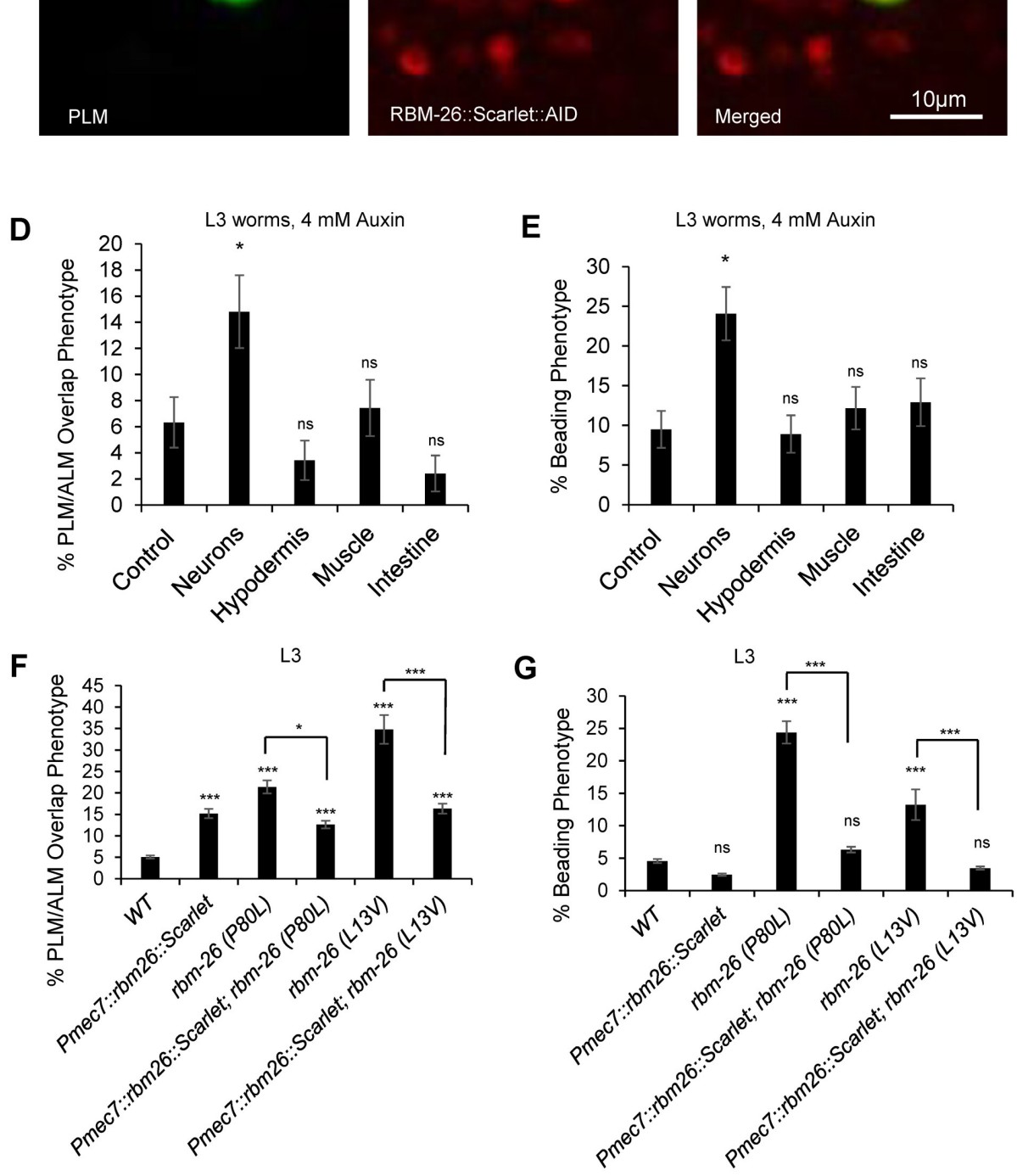

**Fig 3. RBM-26 is expressed in the PLM and functions within neurons to protect against axon degeneration and prevent axon overlap defects.** (A) PLM is identified by GFP expression driven by the *muIs32* transgene that encodes *Pmec-7*::*gfp*. (B) Expression of endogenously tagged RBM-26. The endogenously tagged allele of RBM-26 is *rbm-26 [syb2552(rbm-26*::*Scarlet*::*AID)]*. A and B are merged in (C). (D, E) Neuron-specific degradation of RBM-26::Scarlet::AID causes the PLM/ALM overlap phenotype and axon beading. However, degradation in hypodermis, muscle, and intestine do not cause these phenotypes. Control strain is *rbm-26*::*Scarlet*::*AID*. (F, G) The PLM/ALM overlap and the beading phenotypes caused by mutations in *rbm-26* are rescued by the expression of the *cueSi52(Pmec-7*::*rbm-26*::*scarlet*::*unc-54 3' UTR)*

transgene, which expresses RBM-26 in PLM and other touch receptor neurons. Asterisks indicate statistically significant difference, Z-test for proportions (*$p < 0.05$, ***$p < 0.001$), and error bars represent the standard error of the proportion. $n = 150$ for D and E and 200 in F and G. Touch receptor neurons were visualized in D–G with the *muIs32* transgene. Underlying data can be found in S1 Data.

stage and persists through the L3 stage, when these mutants die (Fig 3F). We also found that all 3 *rbm-26* alleles caused a reduction in the number of mitochondria in the proximal axon (50 μm closest to the cell body; see Fig 4G). However, none of the *rbm-26* alleles affected the number of mitochondria in the distal axon (50 μm closest to the axon tip; see Fig 4H). These observations suggest that RBM-26 is needed to maintain a normal density of mitochondria in the proximal PLM axon.

Excess production of ROS is a key marker for mitochondrial dysfunction [34,35]. Therefore, we asked if the *rbm-26(null)* mutation causes an increase in ROS production by mitochondria. For this experiment, we used mitoTimer, an oxidation sensor that is tethered to mitochondria and changes irreversibly from green to red as it becomes oxidized [36]. We analyzed the mitoTimer red:green ratio in the PLM cell body because the mitoTimer signal in axons was not sufficient for quantitation. We found that the *rbm-26(null)*, *rbm-26(P80L)*, and *rbm-26(L13V)* mutations all increase the mitoTimer red:green ratio in the PLM cell body relative to wild type (Figs 5A–5E and S9), suggesting that loss of RBM-26 function causes mitochondria to produce excessive ROS and that RBM-26 is needed to maintain normal mitochondrial function in neurons. Overall, these results suggest that loss of RBM-26 causes dysfunction of mitochondria in the neuronal cell body, which could lead to a decrease in mitochondria density in the axon.

## RBM-26 binds to the *mals-1* mRNA and negatively regulates its expression

To understand the molecular mechanisms of RBM-26 function, we conducted a screen to identify RNAs that bind to RBM-26. For this screen, we used ultraviolet light to induce RNA-protein crosslinking in strains expressing RBM-26::3XFLAG and in controls expressing GFP::3XFLAG and Scarlet::3XFLAG (see Methods for details). We then used an anti-FLAG antibody to purify the RBM-26::3XFLAG::RNA and Scarlet/GFP::3XFLAG::RNA complexes, followed by RNA-Seq analysis. We identified 63 RNAs that were significantly enriched in RBM-26::3XFLAG pulldowns relative to Scarlet/GFP::3XFLAG controls (Fig 6A and S2 Table). Among these 63 RNAs, we found 26 protein-coding mRNAs, 14 non-coding RNAs, 8 piRNAs, 10 snoRNAs, 1 tRNA, and 1 rRNA.

We focused on the *mals-1* mRNA because it had, by far, the lowest adjusted *p*-value among the RBM-26-binding partners with human orthologs (S2 Table). The *mals-1* mRNA encodes an ortholog of the mammalian MALSU1 mitochondrial ribosome assembly factor [37]. Using qPCR, we found that loss of RBM-26 function causes an increase in *mals-1* mRNA expression in *rbm-26(P80L)* and *rbm-26(L13V)* mutants relative to wild type (Fig 6B). To analyze MALS-1 protein expression, we used CRISPR to add a 3XFLAG tag to MALS-1, thereby creating the *mals-1(cue37)* allele, which expresses the MALS-1::3XFLAG protein. Since the *mals-1* gene is tightly linked to the *rbm-26* gene, we used CRISPR to introduce the P80L mutation into a strain containing the *mals-1(cue37)* mutation, thus creating the *mals-1(cue37) rbm-26(cue42 [P80L])* double mutant. By performing western blotting to analyze expression of MALS-1::3XFLAG in these strains, we found that the *rbm-26(P80L)* mutation substantially enhances expression of the MALS-1::3XFLAG protein relative to wild type (Fig 6C and 6D). We also obtained similar results with the *rbm-26(L13V)* mutation. These observations indicate that RBM-26 negatively regulates the expression of *mals-1* mRNA and MALS-1 protein.

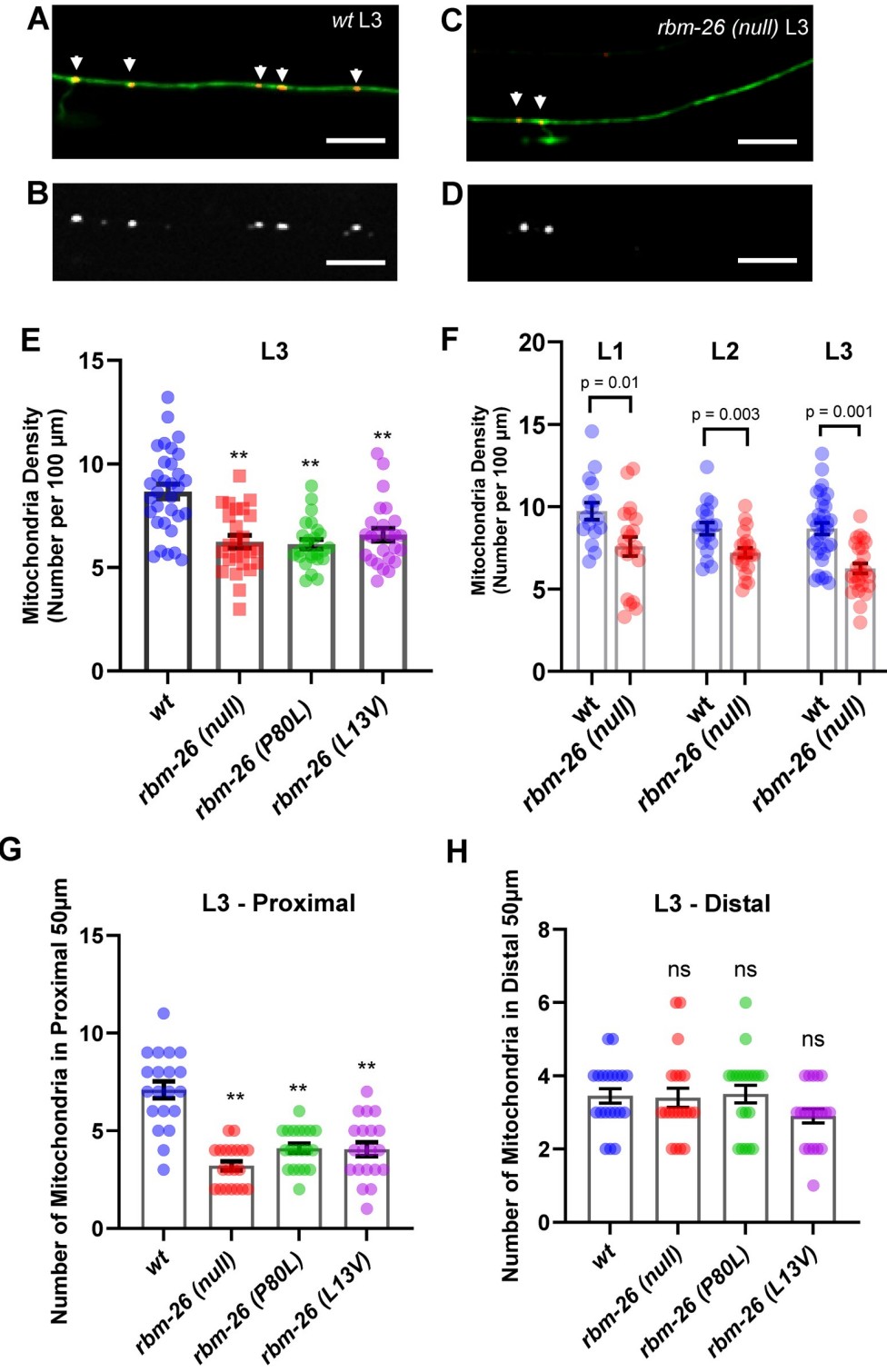

**Fig 4. Loss of RBM-26 function causes a reduction in the density of mitochondria in the PLM axon.** (A–E) RBM-26 loss of function reduces the number of mitochondria in the PLM axon. (A, B) Example of mitochondria (arrowheads) in an L3 wild-type PLM. (C, D) Example of mitochondria in *rbm-26(null)* L3 PLM. PLM axons are visualized with the *muIs32* transgene that encodes *Pmec-7*::*gfp* and mitochondria are visualized with *jsIs1073* transgene that encodes *Pmec-7*::*mito*:*RFP*. B and D show mitochondria only. Images depict the middle of the PLM axon. Scale bars are 10 μm. (E) Quantification of the density of mitochondria (number of mitochondria per 100 μm of PLM axon)

in axons at the L3 stage ($n \geq 25$). Statistical significance was analyzed by one-way ANOVA with a Tukey post hoc test, **$p < 0.01$. (F) Quantification of the density of mitochondria (number of mitochondria per 100 μm) in L1, L2, and L3 PLM axons; 20 PLMs were analyzed for L1 and L2, and 25 PLMs were analyzed for L3. Statistical significance was analyzed by two-tailed Student's *t* test. The reduction in the number of mitochondria in PLM axons in the *rbm-26* mutants was limited to the proximal PLM axon (50 μm closest to the cell body) (G) and was not observed in the distal PLM axon (50 μm closest to the axon tip) (H). A total of 20 PLMs were observed for each genotype in G and H. Statistical significance was analyzed by one-way ANOVA with a Tukey post hoc test, ** $p < 0.01$, ns = not significant. Error bars in E–H are the standard error of mean. Alleles: *rbm-26(null)* is *rbm-26(gk910)*; *rbm-26(P80L)* is *rbm-26 (cue23)*; *rbm-26(L13V)* is *rbm-26(cue34)*. Underlying data can be found in S1 Data.

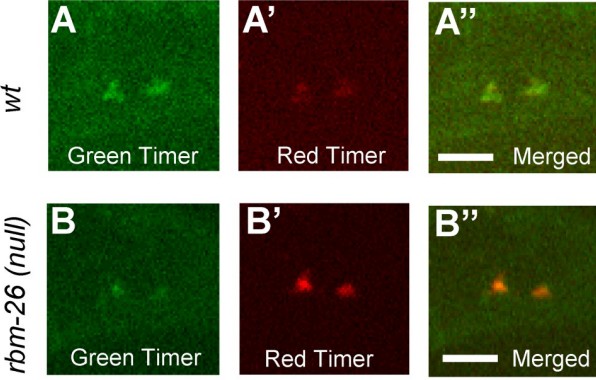

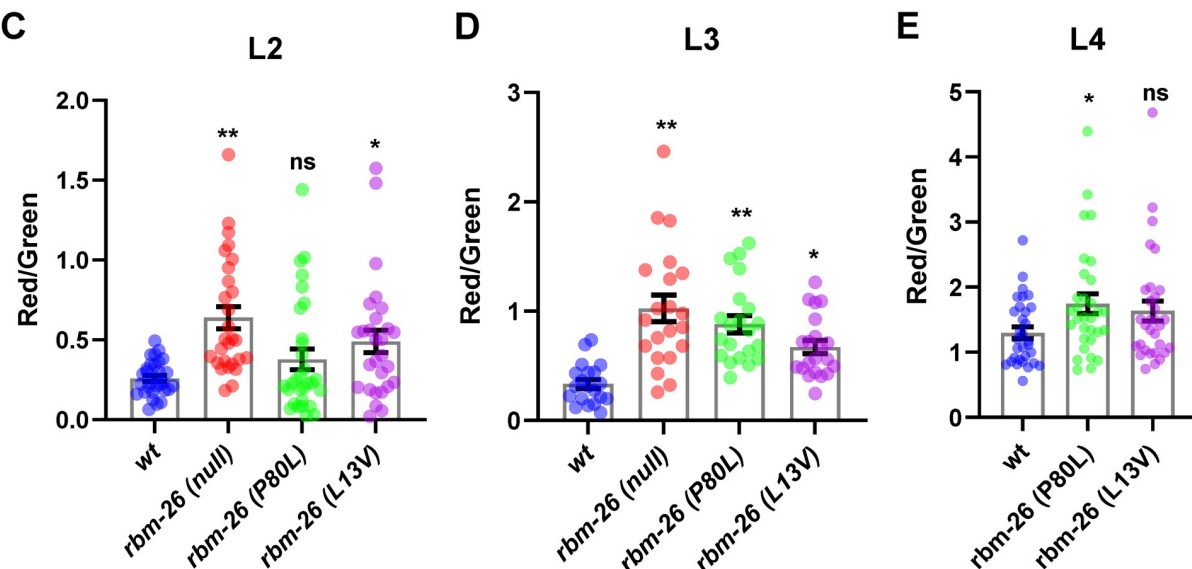

**Fig 5. Loss of RBM-26 causes mitochondria dysfunction in PLM.** RBM-26 loss of function causes excess oxidative activity in mitochondria. To detect oxidative activity in mitochondria, we used mitoTimer, which is a green fluorescent protein when newly synthesized but shifts to red fluorescence when oxidized. (A-A") Example of mitoTimer expression in an L3 wild-type PLM cell body. (B-B") Example of mitoTimer expression in an L3 *rbm-26(null)* mutant PLM cell body. Scale bars are 4 μm. (C–E) Quantification of the ratio of red to green mitoTimer expression in *wt* and *rbm-26* mutant PLM cell bodies in different larval stages. Statistical significance was analyzed with a Tukey post hoc test (*$p < 0.05$, **$p < 0.01$ and ns = not significant); $n = 30$ for C and E, and 21 for D. Error bars represent the standard error of mean. The mitoTimer transgene is *cueSi35 IV (Pmec-7::mitoTimer::tbb-2 3′ UTR)*. Alleles: *rbm-26(null)* is *rbm-26(gk910)*; *rbm-26(P80L)* is *rbm-26(cue23)*; *rbm-26(L13V)* is *rbm-26(cue34)*. Underlying data can be found in S1 Data.

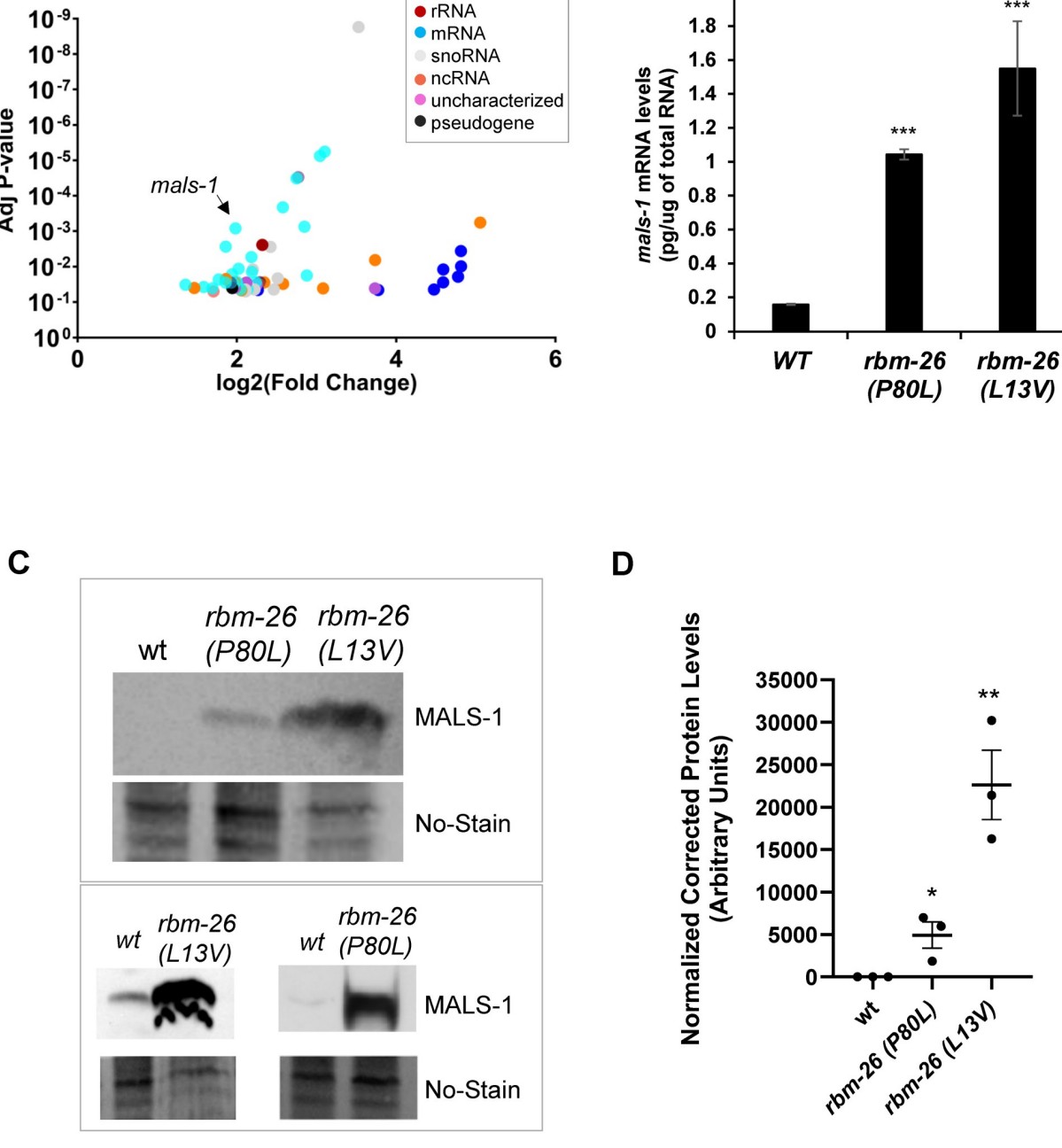

**Fig 6. RBM-26 binds to the *mals-1* mRNA and negatively regulates its expression.** (A) RNA interactors of RBM-26 as identified by UV-RIPseq (see Methods for details). Data generated from 3 biological replicates. (B) Loss of RBM-26 function increases expression of the *mals-1* mRNA. Absolute levels of *mals-1* mRNA quantified using real-time RT-PCR (see Methods for details). Graph represents averages of 3 biological replicates. Error bars are standard deviations. Asterisks indicate statistically significant difference relative to wild type, two-tailed Student's *t* test (***$p < 0.0001$). (C) Loss of RBM-26 function causes an increase in MALS-1 protein expression. Wild-type worms are *mals-1(cue37)* that expresses MALS-1::3X FLAG. CRISPR was used to insert P80L and L13V mutations into the *mals-1(cue37)* chromosome generating *rbm-26 (cue40) mals-1(cue37)* and *rbm-26(cue49) mals-1(cue37)*, respectively; 20 μg of total protein lysate from a mixed stage population was loaded per well in the top panel and specific proteins were detected with an anti-FLAG antibody and enhanced chemiluminescence. Loading was assessed with No-Stain labeling reagent, and 40 μg of total protein lysate was loaded for wild-type and *rbm-26(cue40) mals-1(cue37)* mutants in bottom panels. RIPA buffer was used to prepare lysates used in the top panel and HNM buffer was to prepare lysates used in bottom panel (see Methods for buffer details). A nonspecific band can be seen in *rbm-26 (L13V)* sample in the bottom panel possibly due to buffer conditions, overloading and overexposure. (D) Densitometric quantification of MALS-1 protein levels in western blots. Data are expressed as mean ± SEM from 3 independent experiments. Asterisks indicate statistically significant difference relative to wild type, two-tailed Student's *t* test (**$p < 0.01$, *$p < 0.05$). Underlying data can be found in S1 Data.

 

## Overexpression of MALS-1 phenocopies *rbm-26* loss of function

Since MALSU1 is localized to mitochondria of non-neuronal mammalian cells, we asked if MALS-1 is localized to mitochondria in axons. For this experiment, we created the *cueSi36 IV* transgene, hereafter called *Pmec-7*::*mals-1*::*scarlet* to express MALS-1 in the PLM neuron. Consistent with work on mammalian MALSU1, we found that MALS-1::Scarlet is localized to mitochondria (Fig 7A–7C).

Since we found that loss of RBM-26 function causes an increase in MALS-1 expression and axonal phenotypes, we next asked if overexpression of MALS-1 could phenocopy these axonal phenotypes. For this experiment, we used 2 independent extrachromosomal transgenic arrays to overexpress an untagged version of MALS-1 in the touch receptor neurons. We found that this overexpression of MALS-1 causes an increase in PLM/ALM overlap defects that are significantly greater than those observed in controls (Fig 7D). Overexpression of MALS-1 also caused the axon beading phenotype (Fig 7E). Finally, we found that overexpression of MALS-1 decreases the average density of mitochondria in the PLM axon (Fig 7F). These observations suggest that overexpression of MALS-1 can cause defects that are similar to those caused by loss of RBM-26 function.

## MALS-1 is required for phenotypes caused by loss of RBM-26 function

Since loss of *rbm-26* function causes axonal phenotypes and an increase in MALS-1 expression, we hypothesized that MALS-1 expression is causative for the axonal phenotypes. To test this hypothesis, we asked if *mals-1* is required for the *rbm-26* mutant phenotypes. For these experiments, we used the *mals-1(syb6330)* allele, a near complete deletion of the *mals-1* coding sequence. We also used the *mals-1(tm12122)* mutation, which is a partial deletion of the *mals-1* coding sequence that is predicted to cause early termination of the MALS-1 protein.

To determine if *mals-1* is required for the *rbm-26* mutant phenotypes, we constructed double mutants between alleles of *mals-1* and *rbm-26*. If the *rbm-26* loss of function phenotypes are dependent on *mals-1*, we would expect that the *rbm-26* mutant phenotypes would be suppressed in *rbm-26 mals-1* double mutants relative to *rbm-26* single mutants.

Since the *mals-1* and *rbm-26* genes are tightly linked, we used CRISPR to introduce the P80L mutation into a strain containing the *mals-1(syb6330)* mutation, thereby creating the *rbm-26(cue40 [P80L]) mals-1(syb6330)* double mutant. We found that *rbm-26(cue40 [P80L]) mals-1(syb6330)* double mutants have significantly lower penetrance of axon beading relative to *rbm-26(P80L)* single mutants (Fig 8A). Likewise, *rbm-26(cue40 [P80L]) mals-1(syb6330)* double mutants also have significantly higher mitochondrial density relative to *rbm-26(P80L)* single mutants (Fig 8B). For the PLM/ALM overlap phenotype, we would not expect to see a suppression of the *rbm-26(P80L)* phenotype by *mals-1(syb6330)* because *mals-1(syb6330)* single mutants already have this phenotype at a penetrance equal to *rbm-26(P80L)* mutants, suggesting that loss of *mals-1* function can also cause the axon overlap phenotype. However, we did find that *rbm-26(cue40 [P80L]) mals-1(syb6330)* double mutants had a penetrance of axon termination defects that was equal to *mals-1(syb6330)* single mutants (Fig 8C), consistent with the function of *mals-1* and *rbm-26* in a genetic pathway. We also repeated these experiments with the *rbm-26(L13V)* mutation and found similar results, with the exception that suppression of the beading phenotype by loss of *mals-1* did not quite reach statistical significance ($p = 0.1$). In addition, we noted that the *rbm-26(L13V)* mutation did cause axon overlap defects that were greater than what was observed in the *mals-1(syb6330)* and we found that these defects were suppressed in *rbm-26(L13V) mals-1(syb6330)* double mutants (Fig 8C).

To further test the role of MALS-1 in the *rbm-26* mutant phenotypes, we also tested an independent allele of *mals-1*, *mals-1(tm12122)* and found similar results (Fig 8A–8C). We

 

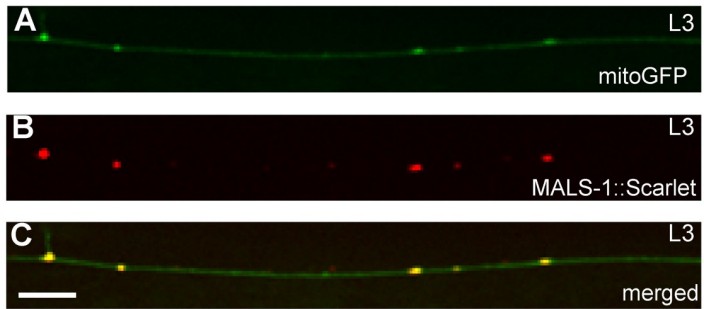

**Fig 7. Overexpression of MALS-1 reduces mitochondria in the PLM axon and causes axon degeneration and axon overlap defects.** (A–C) Overexpressed MALS-1::Scarlet protein colocalizes with mitochondria. (A) Mitochondria visualized with the *jsIs609* transgene, which expresses a mitochondria-targeted GFP (mitoGFP) in the PLM neuron. (B) MALS-1::Scarlet expressed by the *cueSi36 IV* transgene. (C) Merged image showing colocalization of MALS-1::Scarlet with mitoGFP. (D–F) Overexpression of MALS-1 with the *Pmec-7::mals-1* transgenes (*cueEx53* and *cueEx54*) causes defects similar to those caused by loss of RBM-26 function. (D) Axon overlap defects caused by MALS-1 overexpression. (E) PLM axon beading phenotype caused by MALS-1

overexpression. (F) Decrease in mitochondria density (number of mitochondria per 100 μm) in the PLM axon caused by MALS-1 overexpression. Asterisks in panels D and E indicate statistically significant difference relative to wild type, Z-test for proportions (***$p < 0.0001$). For panels D and E, error bars represent the standard error of the proportion. Statistical significance in panel F was analyzed by Student's $t$ test, ***$p < 0.0001$ and error bars represent the standard error of mean. $n = 100$ in D and E, and 20 in F. Underlying data can be found in S1 Data.

noticed that this *mals-1(tm12122)* allele had slightly higher penetrance of defects relative to the *mals-1(syb6330)* mutant and speculate that this might be the result of an incomplete deletion of the sequence coding for MALS-1 in the *tm12122* allele, which might create a truncated protein product that could interfere with the functions of proteins that normally interact with MALS-1. Alternatively, this difference could be caused by a difference in the genetic background. Taken together, these observations indicate that the axonal phenotypes of *rbm-26 (P80L)* and *rbm-26(L13V)* mutants are dependent on MALS-1 function. Moreover, these observations are consistent with the hypothesis that RBM-26 prevents axon degeneration by negatively regulating expression of MALS-1.

We also conducted western blotting as an alternative way to test the interaction between RBM-26, MALS-1, and mitochondria (Fig 8D and 8E). For this experiment, we used the MRPL-58 mitoribosomal protein as a marker for mitochondria. We found that the *rbm-26 (P80L)* mutation causes a significant reduction in the amount of MRPL-58 protein, consistent with a reduction in the number of mitochondria. We also found that this reduction of MRPL-58 expression was suppressed in *rbm-26(P80L) mals-1(syb6330)* double mutants. These observations are consistent with the idea that loss of *rbm-26* causes a reduction in mitochondria that is caused by excessive expression of MALS-1.

## Discussion

Here, we report that RBM-26 functions in neurons to regulate expression of the MALS-1 mitoribosome assembly factor. RBM-26 binds to the *mals-1* mRNA and sharply reduces its expression. Disruptions in this process cause excessive MALS-1 expression, leading to the disruption of mitochondrial function in the developing neuron. These observations provide an example of how posttranscriptional regulation of an mRNA in the nucleus can protect mitochondria in the axon to protect against axon degeneration during neuronal development.

### Relationship between axon overlap defects and axon degeneration phenotypes

We have found that RBM-26 causes defects in both axon overlap and axon degeneration. This is interesting, because the RBM26/27 orthologs of RBM-26 are thought to function with the RNA exosome [17] and variants in the genes that encode RNA exosome components have been implicated in neurodevelopmental syndromes that also cause defects in both axon development and axon degeneration in children [8–10,38,39] (see below for further discussion on the RNA exosome).

Axon overlap defects occur prior to axon degeneration defects in *rbm-26* mutants. Loss of RBM-26 function causes a reduction in mitochondria density that begins during early larval development, coincident with the start of the PLM/ALM overlap phenotype. On the other hand, mitochondrial oxidation is minimal in early larval stages and peaks in later larval stages, coincident with the axon degeneration phenotypes. Overall, these observations suggest that the RBM-26 mutations begin to affect mitochondria in early larval development, giving rise to

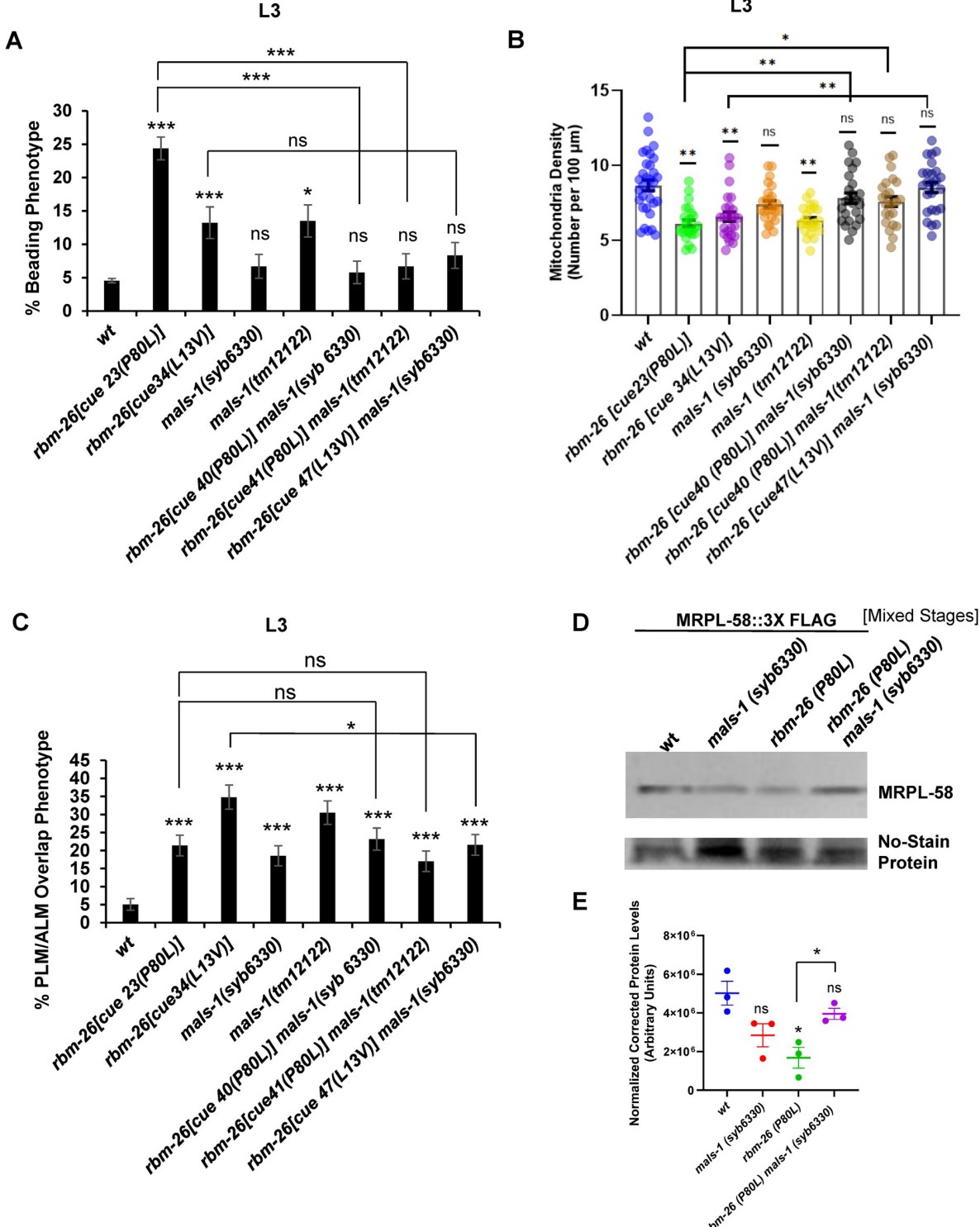

**Fig 8. Loss of MALS-1 function suppresses the mitochondrial and axon degeneration phenotypes that are caused by loss of RBM-26 function.** (A) Loss of *mals-1* function suppresses the PLM axon beading phenotype caused by the *rbm-26* mutations. (B) Loss of *mals-1* function suppresses the reduction of mitochondria caused by the *rbm-26* mutations. (C) Loss of *mals-1* function causes PLM axon termination defect but fails to enhance PLM axon termination defects caused by the *rbm-26* mutations. Control (wild type), *rbm-26 (cue23 [P80L])* and *rbm-26 (cue34[L13V])* data in panels A, B, and C have been reused from Figs 2D, 2F and 5C, respectively. The *rbm-26(cue40 [P80L]) mals-1(syb6330)* double mutant was created by introducing the P80L mutation into the *mals-1(syb6330)* mutant chromosome. The

*rbm-26(cue41 [P80L]) mals-1(tm12122)* double mutant was created by introducing the P80L mutation into the *mals-1(tm12122)* mutant chromosome. The *rbm-26 (cue47 [L13V]) mals-1(syb6330)* double mutant was created by introducing the L13V mutation into the *mals-1 (syb6330)* mutant chromosome. Asterisks in A and C indicate statistically significant difference, Z-test for proportions (*$p < 0.05$, **$p < 0.01$, ***$p < 0.0001$). For panels A and C, error bars represent the standard error of the proportion ($n = 200$). For panel B, error bars represent the standard error of the mean ($n = 25$), and statistical significance was analyzed by one-way ANOVA with a Tukey post hoc test (*$p < 0.05$ and **$p < 0.01$). (D) Loss of RBM-26 protein expression causes a decrease in mitoribosome expression, which was assayed with the MitoRibo-Tag system consisting of the MRPL-58 mitoribosomal protein tagged at its C-terminus with 3XFLAG (MRPL-58::3XFLAG). This MRPL-58::3XFLAG protein was expressed by the *mrpl-58(cue38)* allele, which was created by CRISPR. *mrpl-58(cue38) rbm-26(cue39)* double mutant was created by introducing the P80L mutation into *mrpl-58(cue38)* by CRISPR. CRISPR was also used to insert a 3X FLAG tag at the C-terminus of MRPL-58 in *mals-1(syb6330)* mutant to create *mrpl-58(cue50) mals-1(syb6330)*. Finally, *rbm-26 (P80L)* mutation was inserted into *mrpl-58(cue50) mals-1(syb6330)* to create *rbm-26(cue51 [P80L]) mrpl-58(cue50) mals-1(syb6330)*, and 25 μg of total protein lysate from mixed stage population were loaded per lane and specific proteins were detected with an anti-FLAG antibody and enhanced chemiluminescence. Protein loading was analyzed by No-stain protein labeling reagent. (E) Densitometric quantification of MRPL-58 protein levels in western blots. Data are expressed as mean ± SEM from 3 biological replicates. Asterisks indicate statistically significant difference relative to wild type, Student's *t* test (*$p < 0.05$). Underlying data can be found in S1 Data.

axon overlap defects. In later larval stages, the mitochondrial damage becomes substantially worse, giving rise to axon degeneration.

Axon overlap defects are transient in the P80L and L13V missense mutants. The hypomorphic P80L and L13V missense mutations in RBM-26 cause axon overlap defects that start in the early larval stages and are resolved by the fourth larval stage. By contrast, the *rbm-26* null mutation produces the axon overlap defect at a higher penetrance and this phenotype is not resolved. However, the null mutants die prior to the L4 stage, so we are unable to determine if the phenotype would resolve in later stages. We note that in a prior study, we found that neurodevelopmental disease-associated missense mutations in *unc-16* also cause axon overlap defects that are resolved in later stages in *C. elegans* despite being implicated in severe intellectual disability in humans [40,41]. In this case, *unc-16* null mutations are viable and produce an axon overlap phenotype that persists throughout the life of the worm. On the other hand, a missense mutation in the *egl-19* ortholog of *CACNA1C* produces phenotypes that persist throughout the worm's life [42,43]. Overall, these observations suggest that some, but not all, neurodevelopmental patient-derived mutations can have transient effects in *C. elegans*.

## Potential mechanism for negative regulation of *mals-1* mRNA by RBM-26

Our study demonstrates that RBM-26 is required for the negative regulation of the *mals-1* mRNA and MALS-1 protein but does not address the molecular mechanism through which this happens. Based on prior studies of the RBM26/27 orthologs of RBM-26, we propose that RBM-26 might promote delivery of the *mals-1* mRNA to the nuclear RNA exosome for destruction. This idea is supported by prior work indicating that the orthologs of RBM-26 are part of a highly conserved complex that collaborates with the nuclear RNA exosome to negatively regulate polyadenylated RNA. For example, in human cells, RBM26/27 functions with the PAXT complex, an adaptor that delivers polyadenylated RNA to the nuclear RNA exosome for degradation [17]. Likewise, in *S. pombe*, the Rmn1 ortholog of RBM-26 functions with the MTREC ortholog of the PAXT complex to deliver polyadenylated RNA to the nuclear RNA exosome for degradation [18]. Although the nuclear RNA exosome is primarily responsible for degrading nuclear noncoding RNAs, recent evidence suggests that it can also degrade a select set of mRNAs [17].

## Emerging evidence for the role of the nuclear RNA exosome in protecting against neurodegeneration during infancy

Our study demonstrates that loss of RBM-26 causes axon degeneration during developmental stages in *C. elegans*. Given the role of RBM-26 orthologs with the nuclear RNA exosome,

it is interesting to note that dysfunction in the nuclear RNA exosome has been associated with syndromes that feature neurodegeneration in infants along with defects in neurodevelopment. For example, mutations in the *EXOSC3*, *EXOSC8*, and *EXOSC9* have been implicated in syndromes that include defects in brain development such as hypoplasia of the cerebellum and the corpus callosum [8–10,38,39]. In addition, these syndromes also involve progressive neurodegeneration of spinal neurons that begin in infancy and resemble spinal muscular atrophy (SMA). Consistent with these observations, knockdown of EXOSC3, EXOSC8, and EXOSC9 in zebrafish causes brain malformation and axon pathfinding defects [8,38,44]. Moreover, mutations in EXOSC3 and EXOSC9 are also associated with intellectual disability in humans [10,39] and mutations in EXOSC8 causes behavioral defects in zebrafish [38]. Together, these observations suggest an important role for the RNA exosome complex in protecting against neurodegeneration and preventing neurodevelopmental defects. Together, these observations suggest that the nuclear RNA exosome complex is required to prevent defects in neurodevelopment and protect against neurodegeneration during developmental stages.

## Potential mechanism for how alterations in MALS-1 expression could disrupt mitochondrial function

Our study suggests that loss of RBM-26 function causes a dramatic increase in the expression of the MALS-1 protein that is responsible for a decrease in density and health of mitochondria. However, we do not address the molecular mechanisms through which excessive MALS-1 disrupts mitochondria. Based on prior studies of the MALSU1 ortholog of MALS-1, we propose that excessive expression of MALS-1 could disrupt the assembly of mitoribosomes.

The biogenesis of mitoribosomes is a complex process that involves the assembly of 80 proteins and 2 rRNAs into a large subunit (mtLSU) and a small subunit (mtSSU). This process is facilitated by a large number of mitoribosomal assembly factors including RNA processing enzymes, GTPases, RNA helicases, and kinases. MALS-1 is an ortholog of the human MALSU1 mitoribosomal assembly factor that functions as part of the MALSU1:LOR8F8:mtACP anti-association module [45–47]. This module binds to the mtLSU and blocks premature association of mtLSU with mtSSU. Once assembly of mtLSU is complete, MALSU1:LOR8F8:mtACP is released, thereby allowing association of mtLSU with mtSSU.

Considering the need for MALSU1 to disassociate from mtLSU to allow for final mitoribosomal assembly, we propose that excessive expression of MALS-1 could prevent the dissociation of MALS-1 from the mtLSU, thereby disrupting formation of mitoribosomes. Since mitoribosomes are required for translation of mRNA encoded by mitochondrial DNA, we would expect that disruption of mitoribosomes would cause mitochondrial defects, leading to axon degeneration.

Although our work focuses on mitochondrial and axon defects caused by excessive expression of MALS-1, we also observe decreased mitochondrial density and axon overlap defects in *mals-1* loss of function mutants. These observations are consistent with the role of MALSU1 as a mitoribosomal assembly factor in mammalian cells [37]. Thus, loss of MALSU-1 or MALS-1 expression would be expected to cause defects in mitoribosomal assembly. We have also found that overexpression of RBM-26 can cause axon overlap defects. Although we do not know how overexpression of RBM-26 causes this phenotype, this observation could reflect repression of MALS-1 expression by overexpression of RBM-26. Overall, these observations suggest that mitoribosomal assembly could be disrupted by either loss of MALS-1 or excessive expression of MALS-1.

## Materials and methods

### Strains

The alleles used in this study are listed in S3 Table. The *rbm-26(gk910)* mutation is lethal and was maintained over the hT2 balancer marked with *bli-4* and *(qIs48)*. Worms used for phenotype analysis were maintained at 20˚C on nematode growth medium (NGM) agar plates seeded with OP50 using standard procedures. Worms used for RNA Immunoprecipitation experiments and western blotting were cultured on 8P plates seeded with *E. coli* NA22 strain as food source.

### Genome editing

Genome editing for *mals-1(syb6330)* and *rbm-26*(syb2552 [*rbm-26*::*scarlet*::*AID]*) were performed by SunyBiotech. All other genome editing was performed in our laboratory using a previously described protocol [48,49]. Guide RNAs and homology directed repair (HDR) templates are listed in S4 Table. CRISPR edits were confirmed via PCR and sequencing.

### Transgenes

The *cueSi36 IV* transgene encodes *Pmec7*::*mals-1*::*scarlet*::*tbb-2 3′ UTR*, *cueSi52 IV* transgene encodes *Pmec-7*::*rbm-26*::*scarlet*::*unc-54 3′ UTR*, and the *cueSi35 IV* transgene encodes *Pmec-7*:*mitoTimer*::*tbb-2 3′ UTR*. A single copy of these transgenes was inserted into the *jsTi1493* landing pad on chromosome IV using the Flp recombinase mediated cassette exchange (RMCE) protocol [50]. The *muIs32* transgene is an integrated multicopy array that encodes *Pmec7*::*GFP* and was used to visualize the PLM axon. The *jsIs1073* transgene was obtained from Michael Nonet and encodes *Pmec-7*::*mts*:*tagRFP* and expresses mitochondria-targeted TagRFP (mitoTagRFP). The *jsIs609* transgene was also obtained from Michael Nonet and encodes *Pmec-7*::*mts*::*GFP* and expresses a mitochondria-targeted GFP (mitoGFP) in the PLM neuron. The *cueEx53*(line#1) and *cueEx54*(line#2) transgenes were created by injecting *Pmec-7*::*mals-1*::*tbb2 3′ UTR* at 2 ng/μl + *Podr-1*::*rfp* at 50 ng/μl into worms that express *Pmec-7*::*gfp* and *Pmec-7*::*mitoRFP*.

### Analysis of phenotypes

To quantify phenotypes, worms were mounted on a 5% agarose pad, anesthetized with levamisole, and observed with a 40× objective on a Zeiss Axio Imager M2 microscope as previously described [51–54]. For all experiments, we observed phenotypes in the L3 stage, since the *rbm-26(null)* mutants do not survive beyond this stage. For many experiments, we also included observations at various other stages from L1 to adult. PLM axons were visualized using the *muIs32 (Pmec-7*::*GFP)* transgene. The PLM/ALM overlap phenotype was scored when the PLM axon tip terminated anteriorly to the ALM cell body. We also captured DIC images of worms, fluorescence images of PLM in the same worms and used ImageJ to quantify the length of the worm and corresponding PLMs. Beading refers to focal enlargement or bubble-like lesions which were at least twice the diameter of the axon in size as shown in S3 Fig.

### Imaging

Larval stage images displayed in all figures were obtained by mounting on 5% agarose pad, anesthetizing with levamisole, and imaging with a 60× water objective on a Nikon ECLIPSE Ti microscope equipped with the X-Light V2 L-FOV spinning disk system as previously described [55,56]. NIS-Elements software was used for image acquisition. Image analysis was done with ImageJ or NIS-elements. Identical imaging settings and look-up-tables were used

for control and experimental groups. Imaging for mitochondrial counts was done with a 40× oil objective on a Zeiss Axio Imager M2.

## Analysis of mitoTimer

All images were captured with identical settings on a spinning disk confocal system (see above), and 8-bit images were opened in ImageJ in both red and green channels, the PLM cell body was traced in one channel with the free hand selection tool, and the tracing was restored in the other channel. Raw Integrated Density (RawIntDen) and the area of the region of interest (ROI) were determined in ImageJ. Similarly, RawIntDen and area of a defined background region were obtained. RawIntDen was divided by the area of ROI to obtain total pixel/area. Corrected total pixel/area was obtained by subtracting the background pixel/area from the ROI for both red and green channels.

## Auxin-mediated protein degradation

Tissue-specific degradation of RBM-26::Scarlet::AID was carried out using the auxin-induced degron (AID) system as previously described [33]. Briefly, Naphthaleneacetic acid (K-NAA) (Phytotechnology Laboratories) was dissolved in double distilled water to obtain 250 mM K-NAA solution and filter-sterilized by passing through 25 mm sterile syringe filter (Pall Corporation). Aliquots from sterile 250 mM K-NAA stock was added to autoclaved and slightly cooled (60˚C) NGM to obtain a concentration of 4 mM K-NAA and poured into plates. The next day, 4 mM K-NAA plates were seeded with 250 μl *E. coli* OP50 strain. These plates were used within 2 weeks.

To obtain tissue-specific degradation, homozygous lines were obtained by crossing *rbm-26 (syb2552[rbm-26*::*Scarlet*::*AID])* with the following strains: *reSi1 [col-10p*::*TIR1*::*F2A*:: *mTagBFP2*::*AID*\*::*NLS*::*tbb-2 3′ UTR]* (for hypodermal degradation), *reSi7 [rgef-1p*::*TIR1*:: *F2A*::*mTagBFP2*::*AID*\*::*NLS*::*tbb-2 3' UTR]* (for neuronal degradation), *reSi3 [unc-54p*::*TIR1*:: *F2A*::*mTagBFP2*::*AID*\*::*NLS*::*tbb-2 3' UTR]* (for degradation in muscles), and *reSi12 [ges-1p*:: *TIR1*::*F2A*::*mTagBFP2*::*AID*\*::*NLS*::*tbb-2 3' UTR]* (for degradation in intestine).

L4 stage *C. elegans* of different genotypes were placed on 4mM K-NAA plates, allowed to grow and lay eggs, L3 stage progenies of the parents initially placed on 4 mM K-NAA plates were analyzed for PLM termination defects and PLM beading phenotypes. Knockdown of the RBM-26::Scarlet::AID protein was verified visually. PLMs were visualized with *Pmec-7*::*GFP* transgene. *rbm-26*::*Scarlet*::*AID* without any TIR1 cofactor of AID made up the control group. Knockdown of the RBM-26::Scarlet::AID protein in the PLM was quantified in the same manner as mitoTimer (S10 Fig). To elaborate, ImageJ was used to trace PLM in the green channel and the tracing was restored in the red channel. The RawIntDen was recorded for both channels and subtracted from the background.

## UV-crosslinking RNA immunoprecipitation and RNA-Seq

Worms expressing *wbmIs67* transgene *(eft-3p*::*3X FLAG*::*Scarlet*::*unc-54 3′ UTR)* and *wbmIs72* transgene *(pei-1p*::*3X FLAG*::*GFP*::*unc-54 3′ UTR)* were used as control group. Control and *rbm-26*::*3X FLAG* worms were grown on NGM plates seeded with *E. coli* OP50 strain until they became gravid, then they were washed in M9 (22 mM $KH_2PO_4$; 22 mM $Na_2HPO_4$; 85 mM NaCl; 1 mM $MgSO_4$) and grown for 2 days on 8P plates seeded with *E. coli* strain NA22, and 25,000 mixed stages *C. elegans* were washed 3 times in M9 and placed in an unseeded NGM plate (10 cm) for UV-crosslinking. Plate covers and liquid were removed, and the NGM plates were placed in Spectrolinker X-1000 (Spectronics Corporation) and irradiated with 254 nm UV at energy setting 3000. Worms were then checked

under the microscope to ensure that they were immobilized, following which they were washed in M9, and 100,000 worms were collected in 1 ml of RIPA buffer (50 mM Tris HCl (pH 7.4), 150 mM NaCl, 1% Nonidet NP-40, 0.1% SDS, 0.5% Sodium deoxycholate, Roche cOmplete EDTA-free Protease Inhibitor Cocktail, 25U/ml RNase inhibitor) and homogenized/lysed in BeadBug3 Microtube homogenizer (40 s homogenization with 1 min rest on ice) at 4°C with 3.2 mm diameter chrome steel beads (8 beads per tube). Lysates were placed in fresh tubes, centrifuged at $16,000 \times g$ for 15 min at 4°C. The supernatant was saved as lysate and quantified using Pierce BCA Protein Assay kit, and 100 μl of Pierce Anti-DYKDDDDK (anti-FLAG) magnetic agarose resin was washed 3 times in RIPA buffer and added to 1 ml of protein lysate (2 mg/ml), incubated overnight at 4°C, and anti-flag agarose bound lysate was separated with the help of a magnet while the supernatant was discarded. The agarose resin was washed once with high salt buffer (50 mM Tris HCl (pH 7.4), 1 M NaCl, 1 mM EDTA, 1% Nonidet NP-40, 0.1% SDS, 0.5% Sodium deoxycholate, Roche cOmplete EDTA-free Protease Inhibitor Cocktail and RNase Inhibitor) and 3 times with low salt buffer (20 mM Tris HCl (pH 7.4), 10 mM $MgCl_2$, 0.2% Tween-20, Roche Protease Inhibitor and RNase Inhibitor). To elute RNA, the agarose resin was dissolved in Proteinase K buffer (50 mM Tris-HCl (pH 7.4), 150 mM NaCl, 1 mM $MgCl_2$, 0.05% NP-40, 1% SDS) and incubated with 1.2 mg/ml Proteinase K at 55°C for 30 min with gentle agitation. RNA was extracted with phenol:chloroform and precipitated with 0.25 M Ammonium Acetate, 0.12 M Lithium Chloride, and 85% Ethanol. The pellets were washed with 75% ethanol, dissolved in DEPC-treated water and sent to Azenta US, Inc. for library preparation and RNA sequencing. Raw data obtained from Azenta US, Inc. was analyzed to identify RNA interactors of RBM-26. For RNA-Sequencing pre-processing, adapters were trimmed using Cutadapt [57], and contaminating rRNA reads were removed using bowtie2 [58,59]. Reads were mapped to the Caenorhabditis elegans transcriptome (Caenorhabditis_elegans. WBcel235) using STAR [60] and differential gene analysis was performed using the DESeq2 package [61] in R Studio. Mapped reads were sorted and indexed using samtools [62].

## Absolute quantification of *mals-1* mRNA

The 300 N2 wild-type L4 worms and 300 *rbm-26* mutants [*rbm-26(cue23)* and *rbm-26(cue34)*] L4 worms were collected in Trizol. Following 5 rounds of freeze thaw on dry ice with 1 min of vortexing during thawing, RNA was extracted as per the manufacturer's protocol. We purchased the following synthetic RNA from Azenta US, Inc:GCACGAAGGAGCACAAAAC-GAAGCAAUGGAUGGUAUGUCAGUGAGGUGGAAAGA GUUCAGGUGCACG. The synthetic RNA was diluted in nuclease free water and the following concentrations were confirmed with Qubit RNA Broad Range Assay Kit (Thermo Fisher Scientific): 10 ng/μl, 1 ng/μl, 0.1 ng/μl, 0.01 ng/μl, 0.001 ng/μl, and 0.0001 ng/μl. The diluted RNA standards and 1 μg of total RNA extracted from wild-type and *rbm-26* mutant L4 worms were reverse transcribed with 2 μm reverse primer (CGTGCACCTGAACTCTTTCC) and Superscript First-Strand Synthesis System (Thermo Fisher Scientific) following the manufacturer's protocol. Approximately 1 μl cDNA was used with 10 μl 2× SsoAdvanced Universal SYBR Green Supermix (Biorad) and 5 pmol of the forward *mals-1* primer (TGCACGAAGGAGCACAAAAC) and the reverse *mals-1* primer (CGTGCACCTGAACTCTTTCC). Fluorescence emitted during each cycle was monitored using the CFX96 Touch Real-Timer PCR Detection System (Biorad). Dissociation curve confirmed the presence of a single amplicon. Cycle-to-Threshold (Ct) values of the RNA standards were used to plot a standard curve ($R^2 = 0.995$) (S11 Fig), which was then used to deduce mals-1 mRNA quantity. The data presented here comes from 3 biological replicates.

## Western blotting

Worms were grown on NGM plates seeded with *E. coli OP50* until gravid, washed in M9, and plated onto 8P plates seeded with *E. coli NA22*. A total of 50,000 worms were collected in M9, washed 3 times and M9 was replaced with RIPA buffer (50,000 worms/1 ml RIPA buffer). Worms were lysed in Bead Bug3 Microtube homogenizer at 4°C with 3.2 mm chrome steel beads (8 beads/ml RIPA buffer). Following centrifugation at 16,000 ×g for 15 min at 4°C, supernatant was saved (snapfrozen) as total protein lysate and snap frozen. In some cases, to visualize MALS-1 protein, mixed stage worms were lysed in HNM buffer (50 mM HEPES, 100 mM NaCl, 1 mM MgCl$_2$, 1% Triton X-100, 5% glycerol (pH 7.4)) and lysates were prepared as described before. Total protein lysates were quantified through BCA assay; 20 to 25 μg of total protein lysate was run on 12% SDS-PAGE and transferred to nitrocellulose membrane (0.2 μm pore), and 40 μg of total protein lysate was used for the 4 lower panels in Fig 6C. The nitrocellulose membrane was incubated with No-stain protein labeling reagent for 10 min following the manufacturer's instructions. The membrane was washed 2× in water and 3× in TBST (20 mM Tris, 150 mM NaCl, 0.1% Tween-20). Then, the membrane was blocked in 5% non-fat milk in TBST, probed with primary antibody, washed 3× in TBST, probed with secondary antibody, washed in 3× in TBST, developed with Super Signal West PICO Plus (Thermo Fisher Scientific), and imaged with iBright CL1500 (Thermo Fisher Scientific). The iBright CL 1500 on-instrument analysis software and No-Stain Labeled Membrane setting was used for normalization. iBright analysis software was used for band intensity quantification, which was reevaluated with ImageJ. Antibodies used in this study are listed in S4 Table.

## Supporting information

**S1 Fig. Long exposure of Fig 1C.** Representative western blot of 3 biological replicates showing expression of RBM-26::3XFLAG, RBM-26 P80L::3XFLAG, and RBM-26 L13V::3XFLAG proteins. A total of 20 μg of total protein lysate was loaded per well and specific proteins were detected with an anti-FLAG antibody and enhanced chemiluminescence. Protein loading was quantified by No-Stain Protein Labeling Reagent.
(PDF)

**S2 Fig. RBM-26 is required for proper PLM termination in larval stage L3.** PLM length as a ratio of body length at L3 in wild type (wt), *rbm-26 (null), rbm-26 (P80L), and rbm-26 (L13V)* mutants. Error bars = SD. Statistical significance was analyzed by one-way ANOVA with a Tukey post hoc test, ** $p < 0.01$. n for wt = 24, n for *rbm-26 null* = 21, n for *rbm-26 (P80L)* = 20, and n for *rbm-26 (L13V)* = 22. Alleles: *rbm-26(null)* is *rbm-26(gk910)*; *rbm-26(P80L)* is *rbm-26(cue23)*; *rbm-26(L13V)* is *rbm-26(cue34)*. Underlying data can be found in S2 Data.
(PDF)

**S3 Fig. Examples of degeneration phenotypes.** (A) Example of beading phenotype. Only those focal enlargements along PLM axon that were about twice the diameter of the axon were considered to be beads. (B) Example of blebbing phenotype. (C) Example of waviness in PLM axon. (D) Quantification of the number of beads per 100 μm of PLM axon (bead density) observed in wild type (wt), *rbm-26 (null)*, *rbm-26 (P80L)*, and *rbm-26 (L13V)* at L3. Error bars are standard error of mean. Statistical significance was analyzed by one-way ANOVA with a Tukey post hoc test, * $p < 0.05$ ($n = 28$). Underlying data can be found in S2 Data.
(PDF)

**S4 Fig. Various defects were observed in PVD neuron of *L4 rbm-26 (P80L)*, which were absent or extremely rare in wild type.** (A) Wild-type L4 PVD–primary (1°), secondary (2°),

tertiary (3˚), and quaternary (4˚) dendrites can be seen. (B) Beading can be seen in tertiary dendrites and some quaternary dendrites are missing in *rbm-26 (P80L)* L4 PVD neuron. (C) Quantification of phenotypes observed in *rbm-26 (P80L)* mutants, and 50 wild-type L4 PVD neurons were observed and 60 rbm-26 (P80L) L4 PVD neurons were observed. PVD neurons were visualized by the *wdIs52 [F49H12.4::GFP + unc-119(+)]* transgene, which expresses GFP in PVD. Asterisks indicate statistically significant difference relative to wild type, Z-test for proportions (* $p < 0.05$), and error bars represent the standard error of the proportion. Under-lying data can be found in S2 Data.
(PDF)

**S5 Fig. RBM-26 is expressed in multiple tissues.** RBM-26 is expressed in the nuclei of neu-rons and hypodermal cells. Neurons are identified by the pan-neuronal expression of GFP by the *evIs111* transgene (panels A and D). Hypodermal nuclei are identified by tissue-specific expression of blue fluorescence protein tagged with a nuclear localization signal and an auxin-induced degron *(col-10p::TIR1::F2A::mTagBFP2::AID::NLS::tbb-2 3' UTR)* (panels C and D). Panels B and D show endogenously tagged RBM-26 *(rbm-26::Scarlet::AID)*. Panels A–C are merged in D. "N" in panels A, B, and D indicate neuron and "Hn" in panels B, C, and D indi-cate hypodermal nuclei.
(PDF)

**S6 Fig. RBM-26 is ubiquitously expressed.** Representative Z-stack projection of endogenous RBM-26 protein tagged with wormScarlet (RBM-26::SCARLET::AID) at larval stage L3.
(PDF)

**S7 Fig. AID-related transgenes do not affect axon termination (A) or cause beading pheno-type (B) in the absence of Auxin.** Worms expressing auxin inducible degron (AID) in frame with RBM-26 and tissue-specific TIR-1 as indicated on the X-axis were placed in Auxin free worm plates at L4 and their progeny were observed at L3; "*n*" is between 125 and 150. N.S. = Not Significant (Z-test for proportions). Error bars represent the standard error of the propor-tion. Underlying data can be found in S2 Data.
(PDF)

**S8 Fig. Auxin treatment in the absence of RBM-26::Scarlet::AID does not cause PLM axon termination defect or beading phenotype.** L4 Wild type or Neuronal TIR-1 (without AID tagged RBM-26) were placed on 4 mM synthetic Auxin-coated plates, allowed to lay eggs and their L3 progenies were analyzed for (A) PLM termination defect and (B) beading phenotype; "*n*" is 132 for wt and 138 for Neuronal TIR-1. N.S. = Not Significant (Z-test for proportions). Error bars represent the standard error of the proportion. Underlying data can be found in S2 Data.
(PDF)

**S9 Fig. Example of mitoTimer expression in an L3 wild type [A–C], *rbm-26(P80L)* [D–F] and *rbm-26(L13V)* [E–G] PLM cell body.** mitoTimer is identified by the transgene *cueSi35 (Pmec-7::mitoTimer::tbb-2 3' UTR)*. *rbm-26(P80L)* is *rbm-26(cue23)*; *rbm-26(L13V)* is *rbm-26 (cue34)*. Scale bars are 5 μm.
(PDF)

**S10 Fig. Degron-mediated depletion of RBM-26 in PLM.** L4 *rbm-26 (syb2552); resi7* worms that express RBM-26::Scarlet::AID and F-box Transport Inhibitor Response 1 (TIR-1) in neu-rons were placed in normal NGM plates (A–C) or on plates coated with 4 mM synthetic auxin (D, E), allowed to lay eggs and L3 progenies were analyzed through confocal microscopy. PLM neuron was visualized with the help of the *muIs32 (Pmec-7::gfp)* transgene. Scale bars are

10 μm. Auxin treatment causes depletion of RBM-26 protein, which was quantified in G as a ratio of RBM-26::Scarlet intensity relative to the GFP expression in PLM. Underlying data can be found in S2 Data.
(PDF)

**S11 Fig. Example of RNA standards used to deduce the quantity of *mals-1* mRNA present in 1 μg of total RNA in wild-type and *rbm-26* mutants *(P80L and L13V)* L4 *C. elegans*.** Synthetic RNA corresponding to 68 nucleotides in the last exon and the 3′ UTR of *mals-1* mRNA was purchased from Azenta Inc. The RNA was diluted 1 ng/μl, 0.1 ng/μl, 0.01 ng/μl, 0.001 ng/μl, and 0.0001 ng/μl in nuclease free water. The dilutions were quantified in Qubit (Thermo Fisher Scientific) using RNA Qubit Broad Range kit (Thermo Fischer Scientific); 1, 0.1, 0.01, 0.001, and 0.0001 ng of RNA standard was used in RT-qPCR reaction. Cycle to threshold (Ct) values obtained from RT-qPCR reaction were plotted against the known quantity of RNA used.
(PDF)

**S1 Table. PLM phenotypes observed in larval stages L1 through L4, one- and two-day adults in wild-type, *rbm-26 (null)*, *rbm-26 (P80L)*, and *rbm-26 (L13V)* worms.** Maternally rescued *rbm-26 (null)* worms do not survive past L3. Asterisks indicate statistically significant difference relative to wild type, Z-test for proportions (*** $p < 0.0001$, ** $p < 0.01$, and * $p < 0.05$) while "ns" indicates no significant difference.
(PDF)

**S2 Table. RNA interactors of RBM-26 identified by RNA Immunoprecipitation followed by RNASeq.**
(PDF)

**S3 Table. Strains used in this study.**
(PDF)

**S4 Table. Resource tables.**
(PDF)

**S1 Data. Underlying data for the main figures.**
(XLSX)

**S2 Data. Underlying data for the supplemental figures.**
(XLSX)

**S1 Raw Images. Raw images for all western blots.**
(PDF)

## Acknowledgments

We thank Michael Nonet, Shohei Mitani, and the Caenorhabditis Genetics Center for strains. We also thank Robert Townley for assistance with experiments on the PVD neuron, Claire de la Cova and Mark McBride for sharing equipment, Claire de la Cova for assistance with imaging experiments, and Scott Aoki for help with the UV-RIPseq experiments.

## Author Contributions

**Conceptualization:** Tamjid A. Chowdhury, Christopher C. Quinn.

**Data curation:** David A. Luy, Dorian Farache.

**Formal analysis:** Tamjid A. Chowdhury, David A. Luy, Dorian Farache.

**Funding acquisition:** Tamjid A. Chowdhury, Amy S. Y. Lee, Christopher C. Quinn.

**Investigation:** Tamjid A. Chowdhury, David A. Luy, Garrett Scapellato, Christopher C. Quinn.

**Methodology:** Tamjid A. Chowdhury, Dorian Farache, Amy S. Y. Lee.

**Project administration:** Christopher C. Quinn.

**Supervision:** Tamjid A. Chowdhury, Amy S. Y. Lee, Christopher C. Quinn.

**Writing – original draft:** Tamjid A. Chowdhury, Christopher C. Quinn.

**Writing – review & editing:** Tamjid A. Chowdhury, Amy S. Y. Lee, Christopher C. Quinn.

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
