## [Editor Report · Decision Letter 0]

17 Jun 2024

Dear Dr Quinn, 

Thank you for submitting your manuscript entitled "Autism candidate gene rbm-26 (RBM26/27) regulates MALS-1 to protect against mitochondrial dysfunction and axon degeneration during neurodevelopment." for consideration as a Research Article by PLOS Biology.

Your manuscript has now been evaluated by the PLOS Biology editorial staff as well as by an academic editor with relevant expertise and I am writing to let you know that we would like to send your submission out for external peer review.

Once your full submission is complete, your paper will undergo a series of checks in preparation for peer review. After your manuscript has passed the checks it will be sent out for review. To provide the metadata for your submission, please Login to Editorial Manager (https://www.editorialmanager.com/pbiology) within two working days, i.e. by Jun 19 2024 11:59PM.

Kind regards,

Christian

Christian Schnell, PhD

Senior Editor

PLOS Biology

cschnell@plos.org

---

## [Decision Letter · Decision Letter 1]

5 Jul 2024

Dear Dr Quinn,

Thank you for your patience while we considered your revised manuscript "Autism candidate gene rbm-26 (RBM26/27) regulates MALS-1 to protect against mitochondrial dysfunction and axon degeneration during neurodevelopment." for publication as a Initial Research Submission at PLOS Biology. Your revised study has been evaluated by the PLOS Biology editors, the Academic Editor and the original reviewers.

In light of the reviews, which you will find at the end of this email, we would like to invite you to revise the work to thoroughly address the reviewers' reports.

As you will see below, the reviewers agree that the manuscript is improved but they also say that there are several points that have not been completely addressed. Overall, they say that the figures need to be prepared more carefully and discussion of the results needs to be improved to adequately reflect their experimental data and avoid over-interpretations. Please fully address all concerns, including the ones concerning the rigor of their study.

Given the extent of revision needed, we cannot make a decision about publication until we have seen the revised manuscript and your response to the reviewers' comments. Your revised manuscript is likely to be sent for further evaluation by all or a subset of the reviewers.

**IMPORTANT - SUBMITTING YOUR REVISION**

*Re-submission Checklist*

*Published Peer Review*

*PLOS Data Policy*

*Blot and Gel Data Policy*

Sincerely,

Christian

Christian Schnell, PhD

Senior Editor

PLOS Biology

cschnell@plos.org

REVIEWS:

Reviewer #1: The authors have added some additional controls that improved the manuscript. There are still several issues related to experimental rigor and data integrity.

1. Images altered with black boxes

As noted in the previous review, the image in Fig. 2A has been altered by using a black box to cover the bottom right corner of the image. In their rebuttal letter the authors said they corrected this, but the black box is still there.

2. Protein size markers altered

In the previous submission the major RBM-26 bands were shown to be migrating at ~72 kD and ~50 kD. They are now shown as migrating at >95 kD and ~60 kD.

3. mals-1::Scarlet

The authors do not address the concern from the previous review that the phenotypes caused by mals-1::Scarlet in Fig. 7 (previously Fig. 6) are an artifact of the Scarlet tag. The concern is that the Scarlet tag creates a dominant or toxic product and the phenotypes would not be seen with expression of untagged mals-1. The authors should clarify in the text that the observed defects may be artifacts of the Scarlet tag and not representative of normal mals-1 activity.

4. Discrepancy with mals-1(tm12122)

The authors also do not address the previous concern that the phenotypes for mals-1(tm12122) are stronger than the putative null mals-1(syb6330). In their rebuttal letter they say that "tm12122 could be acting as a dominant negative" to impair function of the wild-type copy, but this strain is homozygous and in any case this would not explain why tm12122 is worse than null. In the text they "speculate that this might be the result of an incomplete deletion of the sequence coding for MALS-1 in the syb12122 [sic] allele" but they need to add that it also may be due to other mutations present in the strain that are not related to mals-1.

5. Axon overlap defects

The authors now show that the axon overlap, which was a focus of the previous version, is actually transient. I am not sure what it means when they say these defects "resolve" by L4 - it is unclear to me if this is by shortening of the axon or some other mechanism. It remains confusing why loss of mals-1 and overexpression of mals-1 both cause this phenotype, while loss of mals-1 suppresses the other rbm-26 phenotypes. In general I am not sure what to make of this transient defect that seems to be caused by many different manipulations of rbm-26 and mals-1. 

6. Incomplete TIR1 effects

The authors state, "We also note that both of these phenotypes occurred at a lower penetrance in these degradation experiments relative to experiments with the rbm-26 mutant alleles, suggesting incomplete degradation of the RBM-26::Scarlet::AID protein." They need to mention that it may also be because rbm-26 acts from other cell types.

7. Speculation

In my view the speculation goes so far beyond the data that it is inappropriate. Examples include:

- " the degenerative and developmental defects caused by loss of rbm-26 function in C. elegans larvae are reminiscent of defects in human infants who carry mutations in various components of the RNA exosome"

- "we speculate that RBM27 may function with the RNA exosome in humans to protect against neurodevelopmental disorders."

- "We speculate that mitochondria in the proximal PLM axon might be more susceptible to damage because of the higher level of axon transport that occurs in the proximal PLM axon"

- "In later larval stages, the mitochondrial damage becomes substantially worse, giving rise to axon degeneration." (Why not that axon degeneration damages mitochondria?)

- The paragraphs on MTREC/PAXT and cerebellar hypoplasia etc. The connection is interesting but there is no experimental footing in this study.

- Similarly the section on mitoribosomes is way beyond what is supported by the data, especially given the authors' rationale for not including key controls ("these experiments cannot be used to determine anything about the mitoribosomes per se, "Given these limitations we have elected not to try additional mitochondrial markers and have also not included additional rbm-26 alleles for this experiment").

Reviewer #2: In the revised manuscript, the authors conducted detailed phenotypic analyses at different larval stages and found that the PLM axon overextension phenotype, which the authors now describe as ALM/PLM overlap, is only transiently observed during L2-L3 stages in the disease-associated rbm-26 mutants. The authors also examined the degeneration phenotypes in the PVD sensory neuron, where disease-associated rbm-26 mutants exhibited statistically insignificant defects in most aspects. 

Overall the authors added substantial new datasets, and significantly improved the rigidity of their work, while new dataset also revealed the effect of rbm-26 mutations to be more transient and subtle than originally described. 

In addition, a number of issues, some of which have been raised in the original reviews, still remain in the current version of the manuscript. 

1. Definition of neurodegenerative phenotypes during neurodevelopment. 

The authors addressed the concerns of both reviewers about mixing developmental and degenerative phenotypes by defining the degenerative phenotype during larval development. However, in the revised manuscript, the authors mix up the terms 'larval development' and 'neurodevelopment' throughout, including their title, which still remains a major cause of confusion. Regarding the PLM/ALM overlap phenotype during L2-L3 stages, the authors called 'developmental defects', which contradicts previous work demonstrating that the PLM neuron completes its axon development in the first larval stage (Gallegos and Bargmann 2004).

2. handling of data that is inconsistent with the authors' model. 

In the original review, I questioned some data that were inconsistent with the authors' model. Unfortunately, the authors only responded to these in their rebuttal without providing additional statements in the result section in their manuscript. The authors should not make arguments just by using datasets that are consistent with their models. Specific datasets are listed below: 

Figure 3F. I did not find authors explaining why rbm-26 overexpression causes axon overlapping phenotype. 

Figure 8A. The authors did not observe suppression of rbm-26 (L13V) by mals-1 mutation, but this was not described nor discussed. 

line 335-6: 'because mals-1(syb6330) single mutants already have this phenotype at a penetrance equal to rbm-26(P80L) mutants.' This should be more carefully explained in the result section as this contradicts the observation of the mals-1 overexpression phenotype in Figure 7. 

Figure 7F. wildtype mitochondria number in this panel is around 6, which is similar to the rbm-26 mutants in Figure 4E. The discrepancy is not explained. 

3. The definition of 'causality'

In the revised manuscript, I still see the authors using the term 'causative' throughout. The authors responded to my previous comment by stating, 'We also note that the titles for both of the cited articles indicate causation.' The authors should carefully read each cited article to understand what previous work did rather than relying on the title of the papers. Examples are listed below:

Line 66, 71: 'Consistent with a causative role for mitochondria dysfunction in neurodevelopmental disorders' As pointed out in the original review, the authors should pay extra attention when referring to the causal effects. All examples the authors listed in this paragraph only suggest correlation not causation. 

line 399-400: 'despite causing severe intellectual disability in humans [41]' The referenced work the authors cited only confirmed the effect of these mutations in C. elegans, similar to this manuscript. Seeing mutant phenotype in C. elegans does not prove the causal effect of these mutations in humans. 

Other comments are listed below:

Line 51 (and many other places): 'Axon tiling - PLM/ALM overlap phenotype' It is confusing to use two terms interchangeably when describing one phenotype. I suggest that the authors use the PLM/ALM overlap phenotype. 

line 188-189: Table S1. The degeneration phenotypes, except for the axon beading phenotype, are very minor, with no statistical analyses provided. 

line 198: 'we also observed the PVD neuron and observed signs of degeneration in both its axon and dendrites' The authors should note that the phenotype is very subtle and most of them are statistically insignificant.

line 295: Figure 6C. Please include the original gel image, as the presented panel does not even include entire MALS-1 protein bands.

line 334: Figure 8B. The color-coding of this panel is confusing. Please use consistent color coding across figures in this paper. The datasets for wild-type, P80L and L13V appear to be identical to the ones in Figure 4E. If the same data were reused, those should be stated in the figure legend.

line 343-4: 'speculate that this might be the result of an incomplete deletion of the sequence coding for MALS-1 in the syb12122 allele.' this is not understandable without reading what the authors described in their rebuttal. The authors should describe in the result or discussion section why they think incomplete deletion in the tm12122 allele causes a more severe phenotype compared to a null mutant. The authors' argument of tm12122 being dominant negative is also misleading as dominant negative cannot be more severe than a null mutant unless there are paralogs of MALS-1.

Line 113-114 'we speculate that RBM27 may function with the RNA exosome in humans to protect against neurodevelopmental disorders.' I would remove this from the introduction as it is purely speculative and has no experiment associated with this.

There are also a number of typos that the authors should carefully check.

---

## [Editor Report · Decision Letter 2]

16 Sep 2024

Dear Dr Quinn,

Thank you for your patience while we considered your revised manuscript "Autism candidate gene rbm-26 (RBM26/27) regulates MALS-1 to protect against mitochondrial dysfunction and axon degeneration during neurodevelopment." for publication as a Research Article at PLOS Biology. This revised version of your manuscript has been evaluated by the PLOS Biology editors and the Academic Editor.

Based on our Academic Editor's assessment of your revision, we are likely to accept this manuscript for publication, provided you satisfactorily address the following data and other policy-related requests:

* We would like to suggest a different title to improve readability/accuracy: 

Autism candidate gene rbm-26 regulates mitoribosomal assembly factor MALS-1 to protect against mitochondrial dysfunction and axon degeneration during neurodevelopment

* Please add the links to the funding agencies in the Financial Disclosure statement in the manuscript details

* DATA POLICY:

Regardless of the method selected, please ensure that you provide the individual numerical values that underlie the summary data displayed in the following figure panels as they are essential for readers to assess your analysis and to reproduce it: 1D, 2DFH, 3DEFG, 4EFGH, 5CDE, 6BD, 7DEF, 8ABCE, S2, S3D, S4C, S7, S8 and S10G

* CODE POLICY

We require the original, uncropped and minimally adjusted images supporting all blot and gel results reported in an article's figures or Supporting Information files. We will require these files before a manuscript can be accepted so please prepare and upload them now. Please carefully read our guidelines for how to prepare and upload this data: https://journals.plos.org/plosbiology/s/figures#loc-blot-and-gel-reporting-requirements

We expect to receive your revised manuscript within two weeks. 

*Published Peer Review History*

*Press*

Sincerely,

Christian

Christian Schnell, PhD

Senior Editor

cschnell@plos.org

PLOS Biology

---

## [Editor Report · Decision Letter 3]

2 Oct 2024

Dear Dr Quinn,

Thank you for the submission of your revised Research Article "Ortholog of autism candidate gene RBM27 regulates mitoribosomal assembly factor MALS-1 to protect against mitochondrial dysfunction and axon degeneration during neurodevelopment." for publication in PLOS Biology. On behalf of my colleagues and the Academic Editor, Mark Alkema, I am pleased to say that we can in principle accept your manuscript for publication, provided you address any remaining formatting and reporting issues. These will be detailed in an email you should receive within 2-3 business days from our colleagues in the journal operations team; no action is required from you until then. Please note that we will not be able to formally accept your manuscript and schedule it for publication until you have completed any requested changes.

PRESS

Sincerely, 

Christian

Christian Schnell, PhD

Senior Editor

PLOS Biology

cschnell@plos.org